# Single-exonuclease nanocircuits reveal the RNA degradation dynamics of PNPase and demonstrate potential for RNA sequencing

Zhiheng Yang [1,2], Wenzhe Liu[2], Lihua Zhao[2], Dongbao Yin[1,2], Jianfei Feng[2], Lidong Li [1] ✉ & Xuefeng Guo [2,3] ✉

The degradation process of RNA is decisive in guaranteeing high-fidelity translation of genetic information in living organisms. However, visualizing the single-base degradation process in real time and deciphering the degradation mechanism at the single-enzyme level remain formidable challenges. Here, we present a reliable in-situ single-PNPase-molecule dynamic electrical detector based on silicon nanowire field-effect transistors with ultra-high temporal resolution. These devices are capable of realizing real-time and label-free monitoring of RNA analog degradation with single-base resolution, including RNA analog binding, single-nucleotide hydrolysis, and single-base movement. We discover a binding event of the enzyme (near the active site) with the nucleoside, offering a further understanding of the RNA degradation mechanism. Relying on systematic analyses of independent reads, approximately 80% accuracy in RNA nucleoside sequencing is achieved in a single testing process. This proof-of-concept sets up a Complementary Metal Oxide Semiconductor (CMOS)-compatible playground for the development of high-throughput detection technologies toward mechanistic exploration and single-molecule sequencing.

RNA degradation is the end of transcriptome information transmission and guarantees high fidelity of genetic information through the degradation of mutational and fragmentary RNA[1,2]. It is also decisive in keeping the healthy operation of the physiological environment by the degradation of the aged non-coding RNA[3,4]. Comprehensive deciphering of the mechanism of RNA degradation is critical for research on the details of the high-fidelity principle for transcriptome information transmission and precise regulation routine of the physiological environment; it also meaningfully promotes the development of technology for accurate sequencing of RNA. Structural information about, and the dynamic properties of, biological macromolecular systems are of fundamental importance for exploring their intrinsic mechanisms and corresponding functions[5-8]. However, traditional macroscopic technologies used to analyze the conformational changes during enzyme–substrate catalytic reactions usually produce collective information about the samples, which can disguise the detailed properties of individual enzymes. To overcome this issue, it is urgently necessary to develop methods that are capable of obtaining intrinsic information at the single-molecule level. The emergence of single-molecule electrical detection provides an advanced way to investigate biological interactions and chemical reactions at the individual event level[9-13]. In comparison with optical technologies, the electrical

[1]State Key Laboratory for Advanced Metals and Materials, School of Materials Science and Engineering, University of Science and Technology Beijing, Beijing 100083, P. R. China. [2]Beijing National Laboratory for Molecular Sciences, National Biomedical Imaging Centre, College of Chemistry and Molecular Engineering, Peking University, 292 Chengfu Road, Haidian District, Beijing 100871, P. R. China. [3]Centre of Single-Molecule Sciences, Institute of Modern Optics, Frontiers Science Centre for New Organic Matter, Tianjin Key Laboratory of Micro-scale Optical Information Science and Technology, College of Electronic Information and Optical Engineering, Nankai University, 38 Tongyan Road, Jinnan District, Tianjin 300350, P. R. China. ✉e-mail: lidong@mater.ustb.edu.cn; guoxf@pku.edu.cn

detection approach can directly record the single-event behavior without fluorescent labeling requirements and bleaching problems; it also possesses higher temporal resolution. These advances in single-molecule electrical measurements have enabled in vitro investigations of single-molecule enzyme dynamics, such as adenosine triphosphatase hydrolysis kinetics[14], the binding mechanism of the DNA polymerase I with deoxyribonucleoside triphosphate analogs[15], peptidoglycan hydrolysis kinetics[16], and DNA binding kinetics of WRKY peptides[17].

Exonucleases degrade polynucleotides in a stepwise manner in either the 3′- to 5′- or 5′- to 3′-direction[18–20]. This has increasingly been integrated into technologies such as nanopores[21], optical tweezers[22], and fluorescence resonance energy transfer[23,24], toward developing strategies for high-throughput RNA-sequencing and understanding the enzymatic mechanism of exonucleases. However, a lack of temporal and spatial resolution significantly restricts the implementation of these technologies to decipher the degradation mechanism of the enzyme polynucleotide phosphorylase (PNPase) at single-base resolution.

*Escherichia coli* PNPase is a typical homotrimeric exonuclease, composed of two *N*-terminal RNase PH domains connected by an α-helical linker, two *C*-terminal domains, and KH/S1 domains. In the presence of phosphoric acids, it can catalyze the step-by-step hydrolysis of RNA, starting from the 3′-terminus, on a millisecond timescale[25–28]. Herein, we build silicon nanowire field-effect transistor (SiNW FET)-based single-protein sensors to realize detection of interactions between individual *E. coli* PNPase molecules and RNA analogs. Integrated with the microsecond temporal resolution technique, this method can directly unveil the detailed degradation process of single-molecule RNAs with high fidelity and single-base resolution in real time. Importantly, using fingerprint mapping analysis, this approach has the capability to efficiently discriminate between different nucleosides in a given RNA sequence, thus constituting the technological basis for future accurate single-molecule sequencing.

## Results

### Designing single-PNPase-modified SiNW FET detectors

To obtain a SiNW FET detector modified with a single PNPase molecule, we adopted a single-molecule modification protocol[29,30]. In brief, the pristine SiNW FET formed by a mechanic-sliding method was spin-coated with a thin polymethyl methacrylate layer. Through precise high-resolution electron-beam lithography and wet etching, a nanoscale special area terminated with Si−H and suitable in size for the conjunction with a single protein molecule was generated on the core-shell SiNWs. After removing the polymethyl methacrylate layer with vast acetone, this special surface of SiNWs was successively modified by undecynic acid hydrosilylation, *N*-hydroxysuccinimide esterification, and maleimide immobilization. Ultimately, by feat of the confinement effect, a single PNPase molecule was conjugated to the molecular bridge on the surface of the SiNW through a thiol−maleimide−Michael addition, relying on a mutated cysteine residue at the bottom of the α-helical domain that has no effect on the activity of the enzyme[31] (Fig. 1a). The details of the device fabrication are provided in Supplementary Information (Supplementary Figs. 1–4). The complete amino acid sequence of PNPase is given in Supplementary Table 1.

In terms of the field-effect theory, the conductance of SiNWs is regulated by their surrounding charge density variation induced by conformational changes of PNPase within the Debye shielding length (Fig. 1b)[32–35]. Consequently, to improve the detection sensitivity, the Debye shielding length ($\lambda_D$) was optimized through regulation of the solution ionic strength (Supplementary Table 2). As is evident, the Debye length increased with the decrement of the ionic strength. Through the dilution of the Tris-buffer (20 mM Tris−HCl, pH 7.6, 30 mM KCl) by 100-fold, the Debye shielding length could be adjusted

to about 11.3 nm[36]. After device fabrication, the conductance−time ($I$−$t$) signal of SiNWs was recorded in real time with a high-speed sampling rate (57,600 Sa•s$^{-1}$) in the blank buffer when a constant source−drain bias voltage (0.3 V) was applied. The same detection was then performed in RNA analog substrate solution, and in RNA analog substrate solution in the presence of H$_3$PO$_4$ (Fig. 1c), respectively. Atomic force microscopy (AFM) and stochastic optical reconstruction microscopy (STORM) were successively used to characterize the single-molecule device. An AFM image showed that an individual PNPase molecule was successfully conjugated on the SiNW surface and the obtained height of PNPase (11.4 nm) was consistent with previous reports[25,27] (~10.6 nm) (Fig. 1d). After injecting an RNA analog substrate labeled with 5′-tetramethylrhodamine azide (TAMRA), a single fluorescent spot was observed above the PNPase by STORM (Fig. 1e), which confirmed the binding interaction of PNPase and the RNA analog labeled with TAMRA.

### Monitoring the PNPase-RNA analog binding process

Using an instrument containing both a super-resolution fluorescence microscope and a high-speed sampling electrical monitoring system, the electrical signal from SiNWs under a 0.3 V bias voltage and the fluorescence intensity of the spot were synchronously recorded. Synchronization in the fluctuation of the bistable electrical and fluorescent signals clarified that the alternation in conductance originated from structural changes on repeated incorporation of an RNA analog [poly(A)$_{30}$] into the PNPase molecule (Fig. 1f), again confirming the formation of a single-protein device. Two conductance states (low and high) can respectively be assigned to the separated (RNA-free) state and the RNA analog-bound state of the enzyme (Figs. 1c and 2a). These results can be explained by a size mismatch in which the aperture of PNPase is smaller than the diameter of RNA analog[31], so the incorporation of RNA analog necessarily expands the structure of the PNPase, as proposed in previous crystallographic studies[25]. In terms of the *p*-type FET, this structural change can enhance the surface electronegativity of PNPase (Fig. 1b), which functions as an additional gate and thus leads to the observed elevation in the device conductance (Figs. 1c and 2a).

To further explore the binding dynamics of PNPase with RNA analog, the temperature dependence was measured at 31 to 39 °C with intervals of 2 °C based on the ultra-temporal electrical monitoring technique (Fig. 2a)[27,37]. Collectively, the dwell time ($\tau$) distributions of both the PNPase-RNA analog-bound structure and the PNPase RNA-free structure showed a single exponential decay function, indicating that each state has a constant average occupancy rate and is an independent stochastic process (see Supplementary Fig. 7). Specifically, both the statistical average dwell time and the occurrence proportion of the PNPase RNA-free structure gradually decreased as the temperature increased, while both the statistical average dwell time and the occurrence proportion of the PNPase-RNA analog-bound structure increased (Fig. 2b, Supplementary Figs. 5 and 6). These results demonstrate that the PNPase-RNA analog-bound structure gradually dominates and becomes more stable with increasing temperature. Through trajectory clustering algorithms[38,39], the datum dots of transformation between the two conductance states were fitted to a continuous curve (Fig. 2c). In order to investigate the dynamic change difference of the PNPase between the RNA analog binding process and the RNA analog releasing process, we calculated the angular velocities ($\omega$) of two adjacent data points. The velocity curves of angular momentum during the transformation process are depicted in Fig. 2c. The contrast values of the full widths at half maxima during the RNA analog binding process (~1.79 ms) and RNA analog releasing process (~2.13 ms) reflected quicker conformational changes of PNPase on RNA analog binding than on RNA analog release.

Intriguingly, as well as the complete transformation between two conformations, a few transient events represented by distinct current

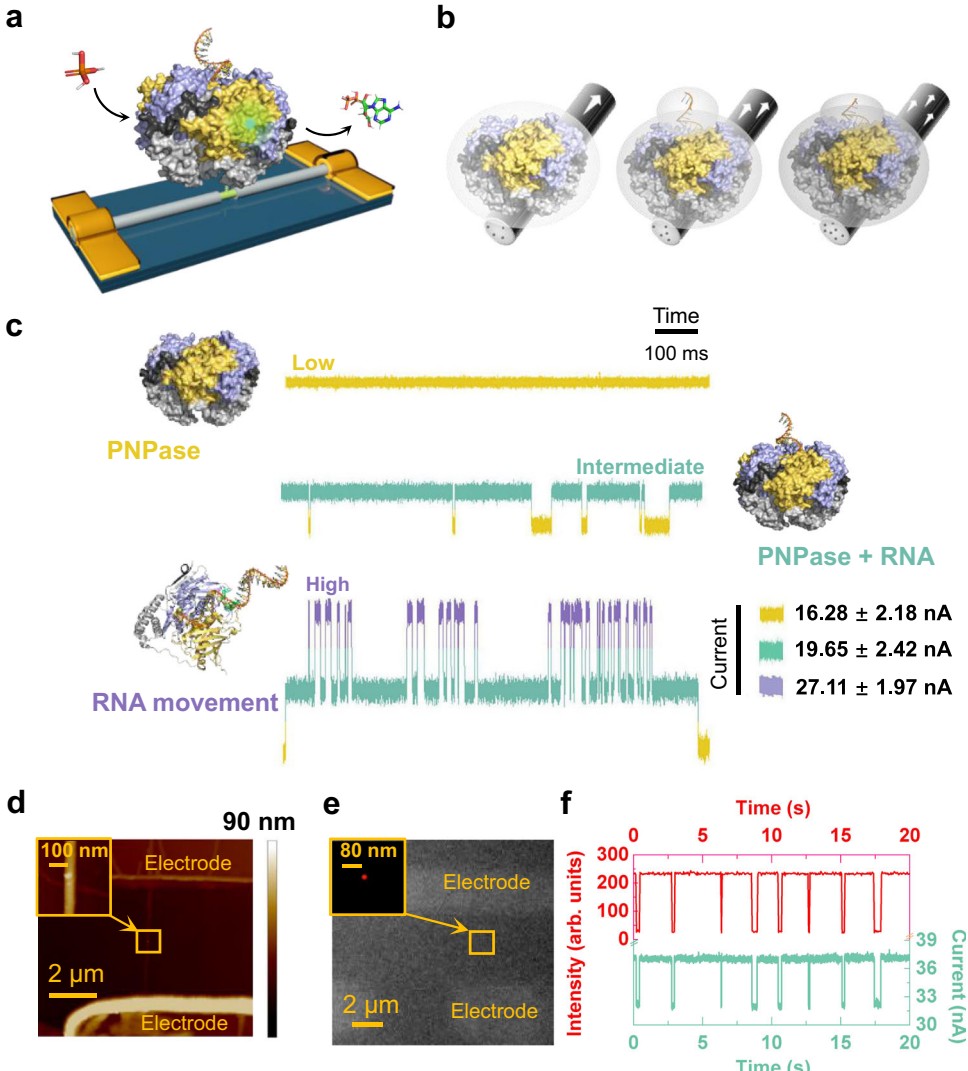

**Fig. 1 | Schematic diagram and characterization of a single-PNPase-molecule-modified silicon nanowire field-effect transistor (SiNW FET). a** Schematic diagram of a SiNW FET device decorated with a single PNPase molecule: RNase PH1 domain, purple; α-helix linker domain, gray; RNase E microdomain, black; RNase PH2 domain, yellow; poly(A)$_{30}$ RNA analog, orange; H$_3$PO$_4$, stick structure; and nucleoside diphosphate, stick structure. Considering the nonconserved properties of the KH and S1 domains, their structures are hidden. **b** Schematic diagram illustrating the effect of PNPase electric density (symbolized with a gray sphere) on the carrier density (black dots) inside the *p*-type silicon nanowire (gray cylinder). The white arrows indicate the orientation of the current. **c** Real-time electrical trajectories of the PNPase structural transformation: before the addition of RNA analog [i.e., poly(A)$_{30}$] (top panel); after the addition of RNA analog [poly(A)$_{30}$] (middle panel); and after further addition of H$_3$PO$_4$ (bottom panel). Bias voltage 300 mV, temperature 37 °C. **d** Atomic force microscopy image of a single PNPase molecule-modified device. Insert: Magnified view of the boxed area, showing a single PNPase molecule immobilized on the surface of the SiNW. The height of PNPase is ~11.4 nm, consistent with previous reports (~10.6 nm). Experiments were repeatedly conducted three times with similar results and one representative image was chosen for analysis. **e** Super-resolution fluorescent image of a single-PNPase-molecule-modified device after the addition of a 5′-TAMRA-labeled RNA analog. Inset: a single fluorescent dot image obtained by stochastic optical reconstruction microscopy. Experiments were repeatedly conducted three times with similar results and one representative image was chosen for analysis. **f** Simultaneously recorded fluorescent and electrical signals during the binding process between PNPase and 5′-TAMRA-labeled RNA analog. arb. units: arbitrary units. Bias voltage 300 mV, temperature 37 °C.

spikes were also captured in real time (Fig. 2d). Based on the biological functions of the KH and S1 domains in the RNA analog binding process (Fig. 2e)[40], "Mode 1" (see Fig. 2d) was speculated to be a backtrack behavior on RNA analog derived from the assistance of the KH and S1 domains before RNA analog release, and "Mode 3" was assumed to be an unsuccessful binding action of PNPase with RNA analog. With the increase of temperature, the event proportions and counts of incomplete transformations both decreased (Fig. 2d). This result indicated that the targeted binding action of PNPase with RNA analog had a temperature dependence.

$\Delta H_T$ and $\Delta S_T$ for the complete transformation from the RNA-free-PNPase structure to the PNPase−RNA analog-bound structure were deduced through linear fitting between $R\ln K_T$ and $1000/T$ (Fig. 2f):

$\Delta H_T = -87.15 \pm 6.10$ kJ•mol$^{-1}$ and $\Delta S_T = -291.56 \pm 19.78$ J•K$^{-1}$•mol$^{-1}$. These values indicate that the structure of PNPase with the RNA analog bound was transformed from a relatively disordered state to a relatively ordered state, and correspondingly, the systemic bond energy of the PNPase-RNA analog-bound structure was converted from low level to high level. We speculate that, on binding of the RNA analog, some stabilizing H-bonds were formed in the PNPase structure. These thermodynamic results are consistent with those from ensemble binding experiments for PNPase with RNA using the isothermal titration calorimetry method[41].

With the increment of the RNA analog dose from 0.2 mM•L$^{-1}$ to 1 mM•L$^{-1}$ at 0.2 mM•L$^{-1}$ intervals, the statistical average dwell time and occurrence proportion of the RNA-free-PNPase structure decreased

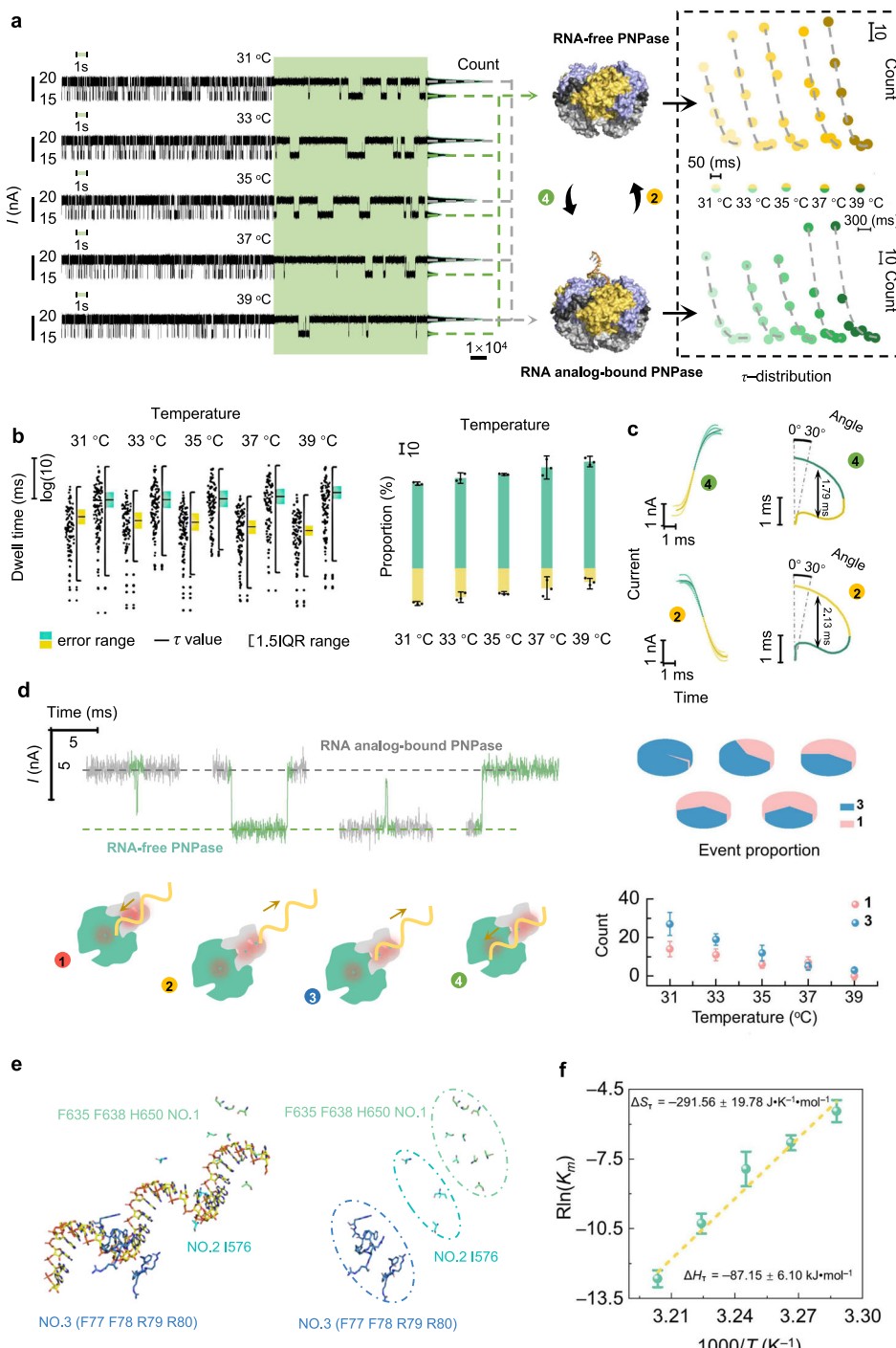

**Fig. 2 | Dynamic analyses of the binding process between PNPase and RNA analog in temperature-dependent experiments. a** Real-time electrical trajectories during the binding process between PNPase and RNA analog. Electrical trajectories marked in green show a magnified 1-s view of each trace and the corresponding current histograms for each trace are shown on the right. The middle panel shows the conversion relation diagram for the RNA-free-PNPase structure (top) and the PNPase–RNA analog-bound structure (bottom). The right-hand panel shows single-exponential distributions of the dwell times corresponding to the two structures (the origin of coordinates: bottom left). **b** Left panel: datum point distributions of the dwell times from one piece of the data and the statistical average dwell time (center) corresponding to each structure. (Below: RNA-free-PNPase; above: RNA analog-bound PNPase). 1.5 IQR range, 1.5-times the interquartile range. Right panel: occurrence proportion distributions of RNA-free-PNPase (yellow) and RNA analog-bound PNPase (green). (two panels mean of *n* = 3 technical replicates from three different single-PNPase-modified SiNW devices, error bars indicate s.d.). **c** Left panel: real-time transformation processes between two conductance states and fitted continuous curves based on trajectory clustering algorithms. Right panel: velocity curves of angular momentum during the transformation process. **d** Left panel: real-time electrical trajectories of entire and incomplete transformation processes (top) and schematic diagrams illustrating the modes of complete and incomplete transformation processes (bottom). Mode 1: backtrack behavior; Mode 2: release process; Mode 3: unsuccessful binding action; Mode 4: complete binding process. Right panel: event proportion (top) and count (bottom) distributions of incomplete transformations. (Mean of *n* = 3 technical replicates from three different single-PNPase-modified SiNW devices, error bars indicate s.d.). **e** Schematic diagrams of the PNPase binding mode with RNA analog (left panel) and the PNPase binding sites to nucleosides and phosphodiester of RNA analog (right panel). No. 1: S1 domain (green); No. 2: KH domain (cyan); and No. 3: RNase PH1 domain (blue). **f** Thermodynamic statistical plots of the binding process between PNPase and RNA analog. (Mean of *n* = 3 technical replicates from three different single-PNPase-modified SiNW devices, error bars indicate s.d.).

(see Supplementary Figs. 5 and 6), while the statistical average dwell time of the PNPase–RNA analog-bound structure was almost unchanged and the occurrence proportion increased gradually, suggesting that the increase of the RNA analog dose improved the binding possibility of PNPase with RNA analog, but did not affect the stability of the PNPase-RNA analog-bound structure (see Supplementary Figs. 8 and 9).

We also individually investigated the effects of pH variation (from 6.4 to 8.0 at 0.4-unit intervals) and $Mg^{2+}$ addition (2 mM·$L^{-1}$) on the structural stability of PNPase bound with RNA analog (see Supplementary Figs. 10–13). During the variation of pH from weak acidity to weak alkalinity, the statistical average dwell time and occurrence proportion of the PNPase-RNA analog-bound structure progressively increased (see Supplementary Figs. 5 and 6), showing that the alkaline condition is favorable for the binding of PNPase with RNA analog and the stability of the PNPase-RNA analog-bound structure. Previous studies[25,42] reported that when RNA is inserted into the active site of PNPase, a loop region of PNPase constructs a multiple conjugation with nucleosides or phosphodiester bonds of RNA analog relying on intermolecular hydrogen bonds or π–π stacking actions. (Fig. 2e). In addition, the KH and S1 domains are both associated with the recognition and binding of RNA analog (Fig. 2e)[25,40]. The improvement of the statistical average dwell time and the occurrence proportion of the PNPase-RNA analog-bound structure could indicate that the stability of the above-mentioned binding sites in the loop region and KH, S1 domains is enhanced in alkaline conditions. In contrast, the statistical average dwell time and the occurrence proportion of the RNA-free-PNPase structure both decreased as the pH increased. Remarkably, the addition of $Mg^{2+}$ improved the occurrence proportion and prolonged statistical average dwell time of the PNPase-RNA analog-bound structure within the pH ranges from 6.4 to 7.6, suggesting that $Mg^{2+}$ accordingly participates in the complexation with phosphodiester bond of analog RNA and amino acid of PNPase in the binding pocket, which was usually found in the substrate binding process of metal catalytic enzyme[25]. Counterintuitively, at pH = 8, the statistical average dwell time and the occurrence proportion of the PNPase-RNA analog-bound structure decreased markedly when $Mg^{2+}$ was added. We speculate that excessive alkalinity might weaken the ability of $Mg^{2+}$ to complex with PNPase and RNA analog (see Supplementary Figs. 5 and 6).

## Monitoring the RNA analog degradation process with single-base resolution

Upon addition of phosphoric acid into the sample chamber covering PNPase-modified SiNWs, a three-level conductance fluctuation appeared (Fig. 3a). The conductance conversion was between an intermediate state and a high state, or between the intermediate state and a low state. This conversion model illustrates that the intermediate state governs the structure-switching route. Based on substrate-controlled and temperature-dependence experiments (see Figs. 1c, 2a, and 3a, discussed above), the intermediate state and the low state can be assigned as the PNPase–RNA analog-bound structure and the PNPase RNA-free structure, respectively. The high-conductance state is speculated to be induced by structural vibration of PNPase associated with the RNA analog ratchet movement on a single-base step close to the active site and the release of nucleoside diphosphate; i.e., the high-conductance state is associated with the RNA analog degradation structure (Fig. 3b). This ratchet movement of RNA might indirectly cause the enhancement of the surface electronegativity of PNPase, which induces a high conductance state. This speculation is entirely consistent with experimental observations—with the substrate poly(A)$_{30}$ we observed exactly 30 fluctuations (the same number of nucleotides as in the RNA analog) between adjacent low-conductance states at 37 °C (Figs. 1c and 3a), clearly demonstrating single-base resolution. A similar mechanical ratchet movement induced by

structural non-equivalence was found for *Caulobacter crescentus* PNPase by X-ray diffraction analysis method[43]. As the temperature was increased in our study, the statistical average dwell time of the intermediate and high states gradually shortened, while the occurrence proportion of the high state increased, which was opposite to the tendency for the intermediate state (Fig. 3c). Through Fourier transform[44,45], the conductance data of the high state were analyzed, which showed that the high state involved a few low-frequency vibrations, whose frequency was increased with increasing temperature, and a few constant high-frequency vibrations belonging to the signal noise (Fig. 3d and Supplementary Fig. 14). Therefore, there were some thermal motions associated with RNA analog movement included in the RNA analog degradation structure. The changes in the Gibbs free energy from the PNPase-RNA analog-bound structure to the RNA analog degradation structure were determined (Fig. 3e); the negative values indicated that the energy of the PNPase system was consumed to drive the single-base movement of RNA analog.

In order to probe the determining factors affecting the single base degradation process, the RNA analog degradation process was systematically examined at different temperatures (31–39 °C at 2 °C intervals) in the presence of $Mg^{2+}$ or $Mn^{2+}$ or $Mg^{2+}$ with $D_2O$, as well as at a $D_2O$ concentration gradient (0%–100% at 20% intervals), and over a pH gradient individually (6.4–8.0 at 0.4-unit intervals) (Fig. 3a and Supplementary Figs. 15–18). The distributions of the statistical average dwell times and the occurrence proportions are shown in Supplementary Figs. 19–25. It was found that both the statistical average dwell time and the occurrence proportion of the intermediate state increased with increasing $D_2O$ concentration, manifesting that the introduction of $D_2O$ slowed the hydrolysis reaction process. Through the calculation of the kinetic isotope effect (KIE) value (3.08 ± 0.37), we confirmed that this reaction process was a primary KIE[46]. The augmentation of the statistical average dwell time and attenuation of the occurrence proportion of the high state hints that $H_2O/D_2O$ plays a key role in the movement of RNA analog (Supplementary Figs. 19 and 20). pH variation from weak acidity to weak alkalinity increased the statistical average dwell time and the occurrence proportion of the intermediate state; the opposite was observed for the low state. This implies that in alkaline conditions, the structure of PNPase favors binding RNA analogs by generating more stable hydrogen bonding between the enzyme and substrate, improving the efficiency of the hydrolysis reaction (Supplementary Fig. 25). Through the counting of the step length in the complete degradation processes (between two adjacent low states) in 20 s, we found that the persistence of single base degradation could be improved by the optimization of testing conditions (Supplementary Fig. 26). Under the weak acidic conditions, the events of the complete degradation process were more frequent, but the degradation products were dominated by small fragments (less than 30 steps). These indicated the whole degradation process of RNA analog was faster under weakly acidic conditions, but the sustainability as a single base step was poor. Unlike pH conditions, the improvement in the temperature did not significantly impact the frequency of the complete degradation process, but mainly contributed to the persistence of the single base degradation.

The statistical average dwell times of the intermediate state and the high state gradually decreased with increasing temperature in the presence of $Mg^{2+}$, $Mn^{2+}$, or $Mg^{2+}$ with $D_2O$ (Fig. 3c and Supplementary Figs. 19 and 20). Unexpectedly, through systematic analyses, a biexponential function could be fitted to the distribution of the dwell time of the intermediate state between two adjacent high states, which indicates that this phase involved two continuous steps (Fig. 3c). Based on the formula:

$$y = (1 + A_1 \times \exp(x_1/- b_2)) \times A_2 \times \exp(x_2/- b_2) + y_0 \qquad (1)$$

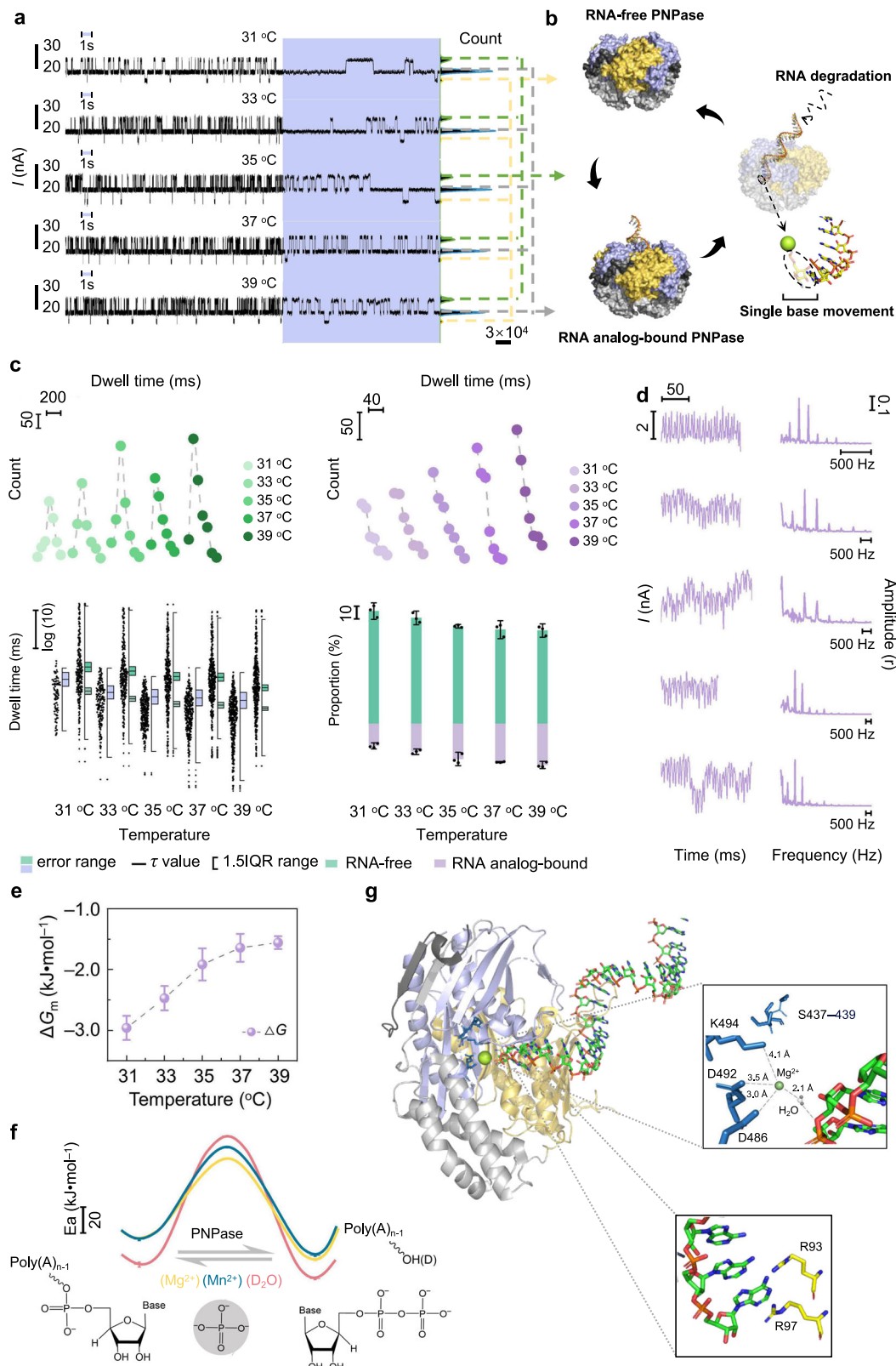

where $A_1$ and $A_2$ are preexponential factors and $b_1$ and $b_2$ are time constants, two statistical average dwell times ($\tau_1$ and $\tau_2$) could be obtained as shown in Supplementary Figs. 21–25. Intriguingly, through control experiments involving various types of homogeneous sequences of the RNA analogs (Supplementary Fig. 27), we found that $\tau_1$ appeared to be almost unchanged, so it can be assigned

to the hydrolysis reaction of phosphate diester bonds, while $\tau_2$ was closely correlated with the sequence, implicating a process associated with the nucleosides of the RNA analog. Based on the Arrhenius formula, the activation energies ($E_a$) of hydrolysis with $Mg^{2+}$ (87.68 ± 2.05 kJ•mol$^{-1}$), $Mn^{2+}$ (105.24 ± 4.21 kJ•mol$^{-1}$), and $D_2O$ (142.17 ± 3.23 kJ•mol$^{-1}$) were obtained through linear fitting between

**Fig. 3 | Dynamic analyses of the degradation process of RNA analog in temperature-dependent experiments. a** Real-time electrical trajectories of the degradation process of poly(A)$_{30}$. Electrical trajectories marked in blue show a magnified 1-s view of each trace and the corresponding current histograms for each trace are presented on the right. **b** Schematic diagram of the conversion between RNA-free-PNPase, RNA analog-bound PNPase, and RNA analog degradation states. **c** Biexponential distributions (upper left) and single-exponential distributions (upper right) of the dwell times corresponding to the RNA analog-bound structure (green) and the degradation structure (purple) (the origin of coordinates: bottom left). Datum point distributions (lower left) of the dwell times from one piece of the data and statistical average dwell time (center) corresponding to each structure (mean of $n = 3$ technical replicates from three different single-PNPase-modified SiNW devices, error range indicate s.d. 1.5IQR range, 1.5-times the interquartile range.). (Below: the degradation structure; above: the RNA analog-bound structure). Occurrence proportion distributions (lower right) of the degradation structure (purple) and the RNA analog-bound structure (green) (mean of $n = 3$ technical

replicates from three different single-PNPase-modified SiNW devices, error bars indicate s.d.). **d** Current trajectories of the degradation structure (left panel) and corresponding illustration of low-frequency vibration peaks analyzed by Fourier transform (right panel). **e** Changes of Gibbs free energy ($\Delta G$) from the RNA analog-bound structure to the degradation structure (mean of $n = 3$ technical replicates from three different single-PNPase-modified SiNW devices, error bars indicate s.d.). **f** Activation energy ($E_a$) distributions for hydrolysis of the phosphodiester bond. Yellow: $Mg^{2+}$ and $H_2O$; red: $Mg^{2+}$ and $D_2O$; blue: $Mn^{2+}$ and $H_2O$. (Mean of $n = 3$ technical replicates from three different single-PNPase-modified SiNW devices, error bars indicate s.d.). **g** Schematic diagram of the PNPase RNA analog-binding mode. The magnified image (top panel) shows the complexation of $Mg^{2+}$ with residues D492, D486, and K494, and the interaction of $Mg^{2+}$ with the phosphodiester bond, relying on a single $H_2O$. Another magnified image (bottom panel) shows the H-bond interaction of the nucleoside at the 3′-terminus with residues R93 and R97.

---

1000/$T$ and $k$ (equal to $1/\tau_1$) (Fig. 3f), which were consistent with the ensemble average values for ester hydrolysis[47]. This result also proved that $\tau_1$ was the dwell time of the hydrolysis reaction.

Accordingly, there might be a binding action near the active site involving (an) amino acid(s) of PNPase and a nucleoside of RNA analog before hydrolysis. A similar binding action was uncovered using the progressive RNA degradation enzymes *C. crescentus* PNPase and archaeal Exosome[43,48]. These interactions seem to be responsible for the proper orientation of RNA analog at the catalytic site, thus allowing the cleavage[49]. Based on the reported Protein Data Bank structure (PDB: 3GME) of PNPase[25], we speculate that during the single-base degradation process, there is a stepwise capture action of nucleosides at the 3′-terminus by binding residues R93 and R97 of PNPase near the reaction center (Fig. 3g)[50].

In terms of the distance between the active site and the RNA analog reactive phosphodiester bond as well as the length of the Mg−O bond in the carboxyl coordination geometry of divalent magnesium hexahydrate[51,52], we hypothesized a model for RNA analog degradation at the PNPase active site (Fig. 3g): On one side of the RNA analog-binding pocket inside PNPase, $Mg^{2+}$ complexes with three $H_2O$ molecules as well as residues K492, D486, and K494 inside the RNase PH1 domain via H-bonds. On the other side, the initial nucleoside at the 3′-terminus contacts with residues R93 and R97 inside the RNase PH2 domain through H-bonds; Subsequently, a single activated coordination $H_2O$ molecule bridges between $Mg^{2+}$ and the phosphodiester bond via H-bonds. Finally, the phosphodiester bond is hydrolyzed with the aid of the captured phosphoric acid bound to S437−439. Similarly, multiple reactive phases of DNA exonuclease were revealed using the fluorescence resonance energy transfer method[24].

### Distinguishing nucleosides between homogeneous RNA analogs

To investigate the degradation dynamics for various nucleotides, different homogeneous sequences−poly(A)$_{30}$, poly(U)$_{30}$, poly(C)$_{30}$, and poly(G)$_{30}$−were respectively analyzed in identical conditions. Representative data are presented in Fig. 4a. Considering that the influence of the specific nucleoside is merely reflected in the coordination with amino acids R93 and R97 through H-bonds, we focused on statistical analyses of the intermediate state (i.e., the RNA analog-bound structure). Dwell time dendrograms for the four sequences are depicted in Fig. 4b (left panel) and fingerprint maps plotted from the density map of the dwell time and current value are shown in Fig. 4b (right panel). Sample data were selected with the prerequisite that the whole degradation process must involve 30 steps, to avoid RNA analog movement missteps and segmentation errors. Because the last hydrolysis reaction happens at the phosphodiester bond between the 29th and 30th nucleotides, the dwell time values of the intermediate state before the 30$^{th}$ site are more valuable for representation of the hydrolysis reaction rate. In this way, the influence of vibration of

the degradation structure without RNA analog movement on statistical analyses is restrained. Taken into the data consolidation of 20 sample volumes, the dwell time dendrograms manifest a good correlation of the dwell time distribution ascribed to the intermediate state on account of a certain sequence, supporting that the dwell time can effectively reflect information about the degradation. Meanwhile, fingerprint maps of the RNA analog-bound structures corresponding to the four types of homogeneous nucleoside sequence exhibit obvious distribution divergencies in the dwell time and current values, confirming the feasibility of distinguishing nucleosides using this technology.

Through biexponential function fitting, we obtained statistical average dwell time values of the bound structures for the four types of homogeneous nucleoside RNA analog (see Supplementary Fig. 27). The statistical average dwell time values $\tau_2$ corresponding to each sequence were: poly(A)$_{30} = 46.33 \pm 3.19$ ms, poly(U)$_{30} = 54.57 \pm 2.99$ ms, poly(C)$_{30} = 34.57 \pm 3.41$ ms, and poly(G)$_{30} = 66.18 \pm 4.07$ ms. $\tau_1$ was almost unchanged for the four sequences [poly(A)$_{30} = 8.86 \pm 0.25$ ms, poly(U)$_{30} = 7.31 \pm 0.32$ ms, poly(C)$_{30} = 6.35 \pm 0.27$ ms, and poly(G)$_{30} = 5.18 \pm 0.24$ ms]. Such dwell time divergences of the binding action for the four nucleosides might originate from steric effects of purines and pyrimidines, as well as the coordination polarity between the nucleoside and positive arginine (R93 and R97).

In order to investigate whether R93 and R97 amino acids participate in the coordination with nucleosides, we mutated R93 and R97 into G93 and G97, which do not have the structure to form hydrogen bonds. The real-time degradation process of the mutant PNPase for poly(A)$_{30}$ was monitored under the same experimental condition (see Supplementary Fig. 28a). Through the fitting for the intermediate state and high state, we found that both states presented a single exponential decay process, which indicated that there was a process lost after mutation (see Supplementary Fig. 28b). The dwell time of the intermediate state ($\tau = 78.84 \pm 4.17$ ms) was augmented in contrast to the value ($\tau_1 = 46.33 \pm 3.19$ ms, $\tau_2 = 8.86 \pm 0.25$ ms) before mutation, showing that the degradation process was delayed after mutation. The reduction of the high-state proportion and the shortening of the step length demonstrated that the stability of single base degradation was also affected (see Supplementary Fig. 28b). After the analysis of the electrical measurement data for the four types of the homogenous sequence, the fingerprints of the intermediate state were presented in Supplementary Fig. 28c. There was not an obvious difference in the fingerprints of the four types of homogenous sequences. In general, these results manifested that R93 and R97 participated in the coordination with the nucleosides of the RNA analog and played a key role in maintaining the stability of a single base degradation.

Thus, by referring to statistical average dwell time values and the distribution discrepancies in fingerprint maps composed of the dwell

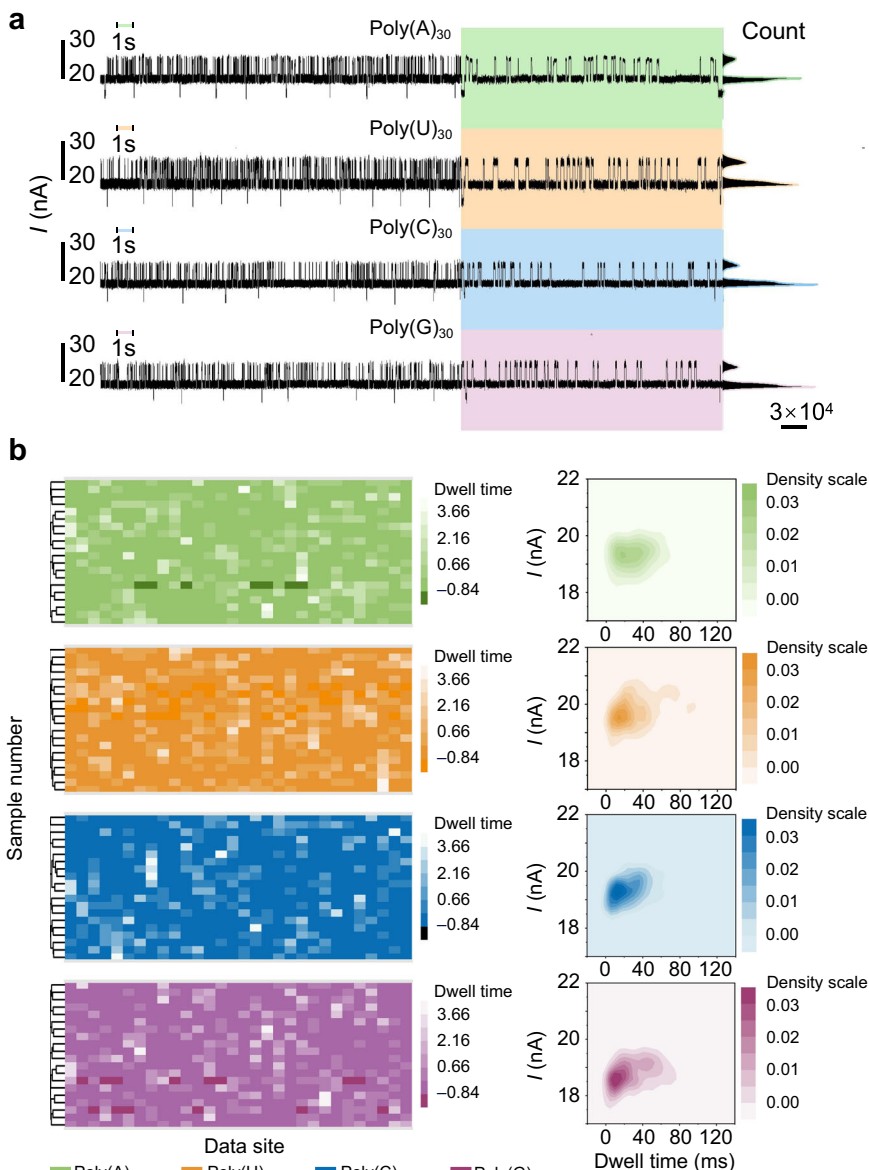

**Fig. 4 | Statistical analyses of real-time electrical trajectories corresponding to four homogeneous RNA analogs. a** Real-time electrical trajectories of the degradation process for poly(A)$_{30}$, poly(U)$_{30}$, poly(C)$_{30}$, and poly(G)$_{30}$ (left panel); 1 mM•L$^{-1}$ RNA analog, pH 7.6, 2 mM•L$^{-1}$ MgCl$_2$, 8 mM•L$^{-1}$ H$_3$PO$_4$, bias voltage 0.3 V, temperature 37 °C. The middle panels (respectively marked in green, bisque, azure, and pink) show the 30-base degradation process of each sequence, again proving single-base resolution. The right panels are the corresponding current histograms for each trace. **b** Dwell time heat maps of the intermediate state for 20 special sample data (involving 30 steps in the degradation process) corresponding to four types of RNA analog sequence (left panel) and dwell time–current density maps for the intermediate state corresponding to each type of RNA analog sequence (right panel).

time and current values, we could efficiently distinguish four kinds of nucleoside sequences. In view of the fact that the nucleoside is the only difference in these four homogeneous sequences, it is reasonable to infer that the specific nucleoside contributes to the stability of the PNPase binding structure with RNA analog.

### Identifying the nucleoside sequence of a heterogeneous RNA analog

Because the binding action is the prerequisite of the hydrolysis reaction, and the statistical average dwell time of the binding action is more significant than that of the hydrolysis reaction (see above), the coordination action of the nucleoside at the 3′- terminus of the RNA analog with residues R93 and R97 of PNPase determines the rate of the single-base degradation step. Considering the dwell time divergences of the coordination action in the different hydrogen-bonding modes of the

four types of nucleosides with R93 and R97 (Fig. 5a, b), which are visually reflected in fingerprint maps, we attempted to measure the binding and degradation of the randomly artificial heterogeneous RNA sequence 5′-CGAUCUUCAUUGCCAAGCGGCUAG CUCAAA-3′ to manifest the single base resolution and the sequence was arranged to avoid the secondary structure barrier. The circus image of the special data (20 samples), where each sample involved 30 single-base degradation steps, was shown in Fig. 5c. It includes a scatter distribution of the current in the middle circle and a distribution of the dwell time in the inner circle. The centralized distribution of the current value and the dwell time reflected the stability and reliability of the detected data. Then, given the heterogeneity of the sequence, we analyzed the dwell times and current values of the intermediate states, taking into account 150 pieces of special data (i.e., 150 RNA degradation data), where each sample involved 30 single-base degradation steps to obtain a collection

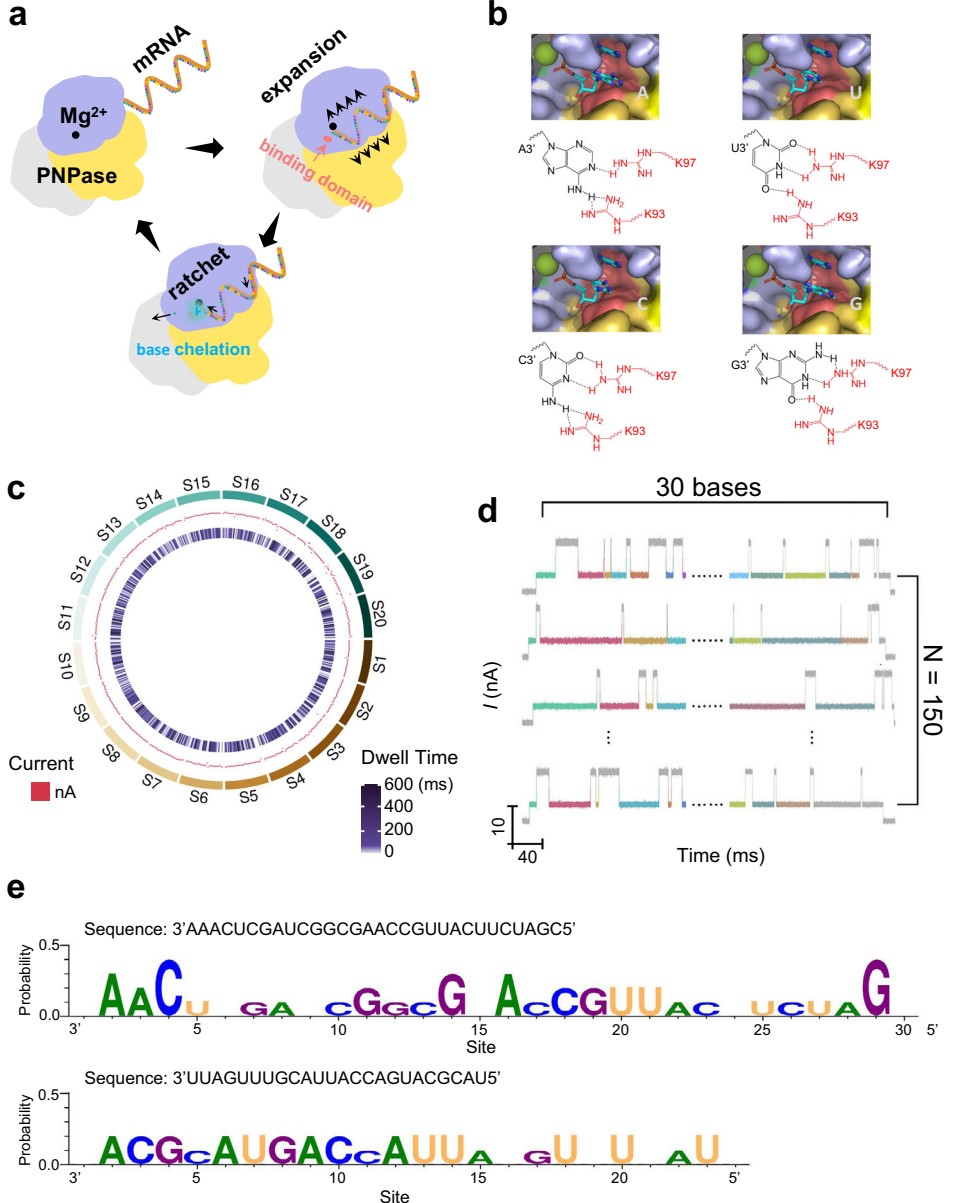

**Fig. 5 | Statistical analyses of real-time electrical trajectories for a heterogeneous RNA analog. a** Schematic illustration of the degradation process of heterogeneous RNA analog. RNase PH1 domain, purple; α-helix linker domain, gray; RNase PH2 domain, yellow; the heterogeneous RNA analog, orange; $Mg^{2+}$ at the active site, black; H-bonding domain of PNPase with the nucleoside at the 3'-terminus of the RNA analog, orange; nucleoside chelation behavior with the binding domain of PNPase at the 3'-terminus, azure. **b** H-bonding modes for four the kinds of nucleoside (i.e., AMP, UMP, CMP, and GMP) at the 3'-terminus of RNA analog with residues R93 and R97 of PNPase. **c** Representative circus diagram of testing data for a heterogeneous RNA analog (20 samples from S1–S20). Red dots represent the current value and the heap map represents the distribution of dwell times. **d** Representative analyses of 150 samples for heterogeneous RNA analog. Each sample involves degradation 30 steps. The same color of the intermediate state in various data samples represents the same location of the nucleoside in the sequence of heterogeneous RNA analog. **e** Identification seq-logo diagrams of the artificially designed sequence (up panel) and the *mccA* gene sequence (bottom panel) in 10 collections (A, adenine; U, uracil; C, cytosine; G, guanine; N, non-matched site). The probability (character size) of every site was obtained by the ratio of the number of the matched bases to the sample volume (i.e., 10).

involving fingerprints of every step from the 5' terminus of the sequence (Fig. 5d). Supplementary Figs. 29–38 show 10 groups of different map collections where everyone reflects the detailed fingerprint maps of 30 sites in the sequence. Relying on comparison with the fingerprint maps ascribed to the four types of nucleosides (Fig. 4b right), nucleoside information for most types in the 5'-CGAUCUUCAUU GCCAAGCGGCUAGCUCAAA-3' sequence could be identified—indeed, through ordinal contrast of these fingerprint profiles, 24 nucleosides could be identified effectively (corresponding to about 80% accuracy in the 30-nucleoside sequence) (Fig. 5e and Supplementary Fig. 39).

In addition, for a more realistic sequencing for transcriptome information relying on this technique, we synthesized a cellular RNA sequence 5'-UACGCAUGACCAUUA CGUUUGAUU-3' (i.e., *mccA* gene), which is responsible for encoding an oligopeptide of seven amino acids corresponding to the primary structure of microcin C7 in *E. coli* cells[53,54]. The potential secondary structures were shown in Supplementary Fig. 40. The RNA substrates were kept warm at 65 °C and then quickly cooled to obtain a few unfold sequences to avoid the repellency effect of the PNPase on the changeable secondary RNA[55]. After electrical measurements and data analysis, the sequence analysis

collections were shown in Supplementary Figs. 41–50. Finally, the identified efficiency of the *mccA* gene sequence could reach about 79.17% (Fig. 5e and Supplementary Fig. 51). Through the electrical measurement for the mixed sample involving an artificially designed sequence (30 nt in length) and a *mccA* gene sequence (24 nt in length), we found that the complete single base degradation processes were mainly characterized by thirty steps (marked with blue) and 24 steps (marked with brown) as shown in Supplement Fig. 52. The statistical distributions of the step length were also shown as located around sites at 30 nt and 24 nt in length (Supplementary Fig. 53). These results demonstrated the single-molecule precision of this electrical testing technique. Collectively, these results demonstrate the promising capability of single-exonuclease nanocircuits in distinguishing different nucleosides at single-base resolution.

## Discussion

In summary, taking advantage of high temporal resolution, sensitivity, and good biocompatibility, we developed a universal methodology using a SiNW FET-based single-molecule device to realize in-depth investigation of RNA analog degradation dynamics in real time. Statistical analysis fully elicited the degradation process at the single-molecule level, including RNA analog binding, single-nucleotide hydrolysis, and single-base movement. Statistical calculation verified the fact that conformational entropy of PNPase drove the binding action with RNA analog. Importantly, a coordination action before hydrolysis, between the nucleoside at the 3′-terminus of the RNA analog and residues R93/R97 in the active site domain of PNPase, was discovered, and is key to the degradation behavior. By drawing fingerprint maps of every site in an RNA analog sequence, we are able to distinguish nucleosides with ~80% (artificially designed sequence) and ~79.17% (*mccA* gene sequence) accuracy at single-base resolution. To achieve practical real-time sequencing of RNA, robust and accurate machine learning algorithms for in-depth data mining discrepancy of nucleosides and genetic engineering technologies for mutating amino acids in the RNA analog-binding pocket to prolong the duration of coordination are needed. It will also be important to amplify the signal differences of different nucleosides. In addition, limited by the natural property of the PNPase, which is suitable for the degradation of short-piece RNAs, the current approach has not realized the identification of longer transcripts[55]. In the future, more practical enzymes and robust analysis algorithms are urgently needed in the realization of sequencing longer transcripts. This single-molecule detection methodology has created an advanced way to investigate the fundamental mechanisms of enzymes, and is naturally applicable to broad DNA, RNA, and peptide identification. This is a profound step in realizing a low-cost de novo sequencing method with the ultimate sensitivity limit.

## Methods

### Device fabrication and characterization

**The device fabrication process.** Silicon wafers with a 1000 nm–thick thermally grown oxide layer were selected as growth substrates and gold nanoparticles with an average diameter of 20 nm (Ted Pella) were chosen as catalysts. 2.5 sccm disilane (Matheson Gas Products, 99.998% Purity) was used as the reactant source and 0.11 sccm diborane (100 ppm, diluted in $H_2$) was used as the *p*-type dopant. Under the condition of a B/Si ratio of 1/1,00,000 and 7.5 sccm $H_2$ as the carrier gas, boron-doped *p*-type SiNWs were synthesized at the temperature of 465 °C for about 25 min.

SiNWs were transferred to the region designed for electrode patterning on the surface of silicon wafers. Through the photolithographic technique, a positive resist (ARP 5350) stripe was generated to retain SiNWs between the electrodes and remove redundant SiNWs by sonication. After that, the resist was washed with acetone. To generate better Ohmic electrical contacts with metal electrodes, the buffered HF solution (40% NH$_4$F:40% HF, 7:1) was used to remove the oxide shell

of the nanowires. Then, 8 nm Cr and 80 nm Au were successively deposited on the silicon wafer by thermal evaporation (ZHD-300, Beijing Technol Science) to form metal electrodes. Subsequently, through electron beam thermal evaporation (TEMD-600, Beijing Technol Science), another 50 nm thick $SiO_2$ protective layer was deposited to passivate the contact interface of the gold electrodes. Finally, the photoresist as well as the $SiO_2$ protective layer between the electrodes was removed by copious acetone to obtain SiNW FET arrays. Considering the testing condition that the drain current was measured in the buffer solution, a negative resist (SU-8, 2002) was utilized to open a window by photolithography to expose the intact surfaces of silicon wires for following modifications and protect the majority of the surface to prevent a generation of the electrical leakage. The windows between electrodes also functioned as a reactive chamber for the measurement of the enzymatic reaction process in vitro. Through systemic treatments, the high–density SiNW FET arrays were obtained as shown in Supplementary Figs. 1 and 2.

**Electrical characterization of SiNW FETs.** Using the heavily doped Si substrate as the global back gate, the electrical characterization of SiNW FETs was conducted at room temperature in the ambient environment with an apparatus having an Agilent 4155 C semiconductor analyzer and a Karl Süss (PM5) manual probe station. SiNW transistors showed the typical *p*-type behaviors with good Ohmic contacts (Supplementary Fig. 3)[35].

### The immobilization process of a single *E. coli* PNPase protein

**Confinement area locating.** A bare SiNW FET device was covered with a polymethyl methacrylate (PMMA 950, A4) layer by spin coating and then baked for about 2 min at 180 °C. By precise high-resolution electron beam lithography with a designed CAD file that includes a -10 nm-wide solid line, a nanogap precursor was generated at the potential position. After that, through sonication at 4 °C for about 1 min, the resist was developed in a mixture of water/isopropanol (V:V = 1:3) for lift-off. As a result, a certain nanogap suitable for the coverage by a single protein in size was generated on the surface of the SiNWs. After the device was washed with deionized water and dried with a $N_2$ gas stream, the device was immersed in a buffered HF solution (HF (40%): NH$_4$F (40%) = 1:7) for about 7 s to remove the amorphous $SiO_2$ shell and expose the silicon surface that is terminated with Si–H bonds. Finally, the PMMA layer was cleaned with acetone.

**Single PNPase protein immobilization.** To realize the successful immobilization of PNPase at the single-molecule level, the hydrosilylation of Si–H bonds was chosen as the initial treatment. In order to improve the yield of alkyne hydrosilylation, we used high-purity undecynic acid as the grafted reagent[17]. The freshly etched device was placed inside the Schlenk bottle and then 3 mg powders of undecynic acid were covered on the device. After 10 h of heating at 90 °C under an argon atmosphere, the device was immersed in dichloromethane. After about 30 s sonication to remove unreacted resides on the surface of the device, the device was dried with a stream of $N_2$ gas. Then, the device was soaked into a mixture solution containing N-hydroxysuccinimide (NHS) (20 mM) and 1-ethyl-3-(3-dimethylaminopropyl) carbodiimide (EDC) (10 mM) for 2 h at room temperature to realize esterification of carboxyl group with NHS (pH = 6.5). The device was thoroughly washed with deionized water and subsequently dried with a stream of $N_2$ gas. The dried device was immediately immersed into a N, N-Dimethylformamide (DMF) solution containing N-(2-Aminoethyl) maleimide Hydrochloride (18 mM) to generate an efficient reaction between activated carboxyl group and maleimide. Then, the device was washed thoroughly with DMF and dried with a stream of $N_2$ gas. Finally, a 50 μL PNPase solution (200 nM PNPase in PBS buffer containing 20 mM Tris-HCl, pH 7.6, 100 mM KCl and 1 mM MgCl$_2$) was added to cover the device surface for 24 h at 4 °C. After the sufficient

reaction, the device was washed with buffer solution as well as weak ultrasonic treatment to remove redundant PNPase (Supplementary Fig. 4).

## Optical characterization

A buffer liquid layer containing a fluorescent-labeled RNA analog substrate was added to cover the surface of the PNPase-modified SiNW FET device (1.8 cm × 3.5 cm) and then attached with a coverslip. Subsequently, the device was placed on the microscope objective stage. The target electrode pair were caught by two extension probes, which were appended to the signal source and drain sockets of the UHFLI lock-in amplifier, respectively. A Nikon Ni–E microscope with a ×100 objective lens was positioned in close contact with the coverslip on the device by feat of the lens oil. The signal collection area of the stochastic optical reconstruction microscopy was zoomed on the modified portion of the single silicon wire. Then, a 548 nm laser was employed to excite the device and an EMCCD (Andor) was used to receive the feedback emitted light and the fluorescence spectra with a 50 ms exposure resolution. The process control cable of a stochastic optical reconstruction microscopy was connected to the trigger terminal of the UHFLI lock-in amplifier (high-speed sampling rate: 57,600 Sa•s$^{-1}$) to provide synchronous triggering. Once the trigger was switched, the fluorescent and electronic signals of the target location were simultaneously recorded within 5 min. After that, through the reconstruction and analysis of the optical pictures using the Advanced Research software, a single fluorescence spot and an enlarged image without the background were merged[17].

## PNPase expression and purification

Protein expression experiments were delegated to Shanghai Sangon Biotech (Shanghai) Co., Ltd. The gene encoding *E. coli* PNPase was produced by whole gene synthesis and subcloned into expression vector pET28a. The resulting plasmid was transformed into *E. coli* cells. Single colonies were isolated and cultured in a liquid medium containing 30 μg/mL kanamycin at 37 °C. When the OD value reached 0.6, 0.5 mM isopropyl β-D-1-thiogalactopyranoside was added. Cells were left to grow overnight at 20 °C and then for 6 h at 37 °C. Bacteria were collected by centrifugation (10,112×*g* for 10 min) and resuspended in buffer (50 mM Tris, 300 mM NaCl, 0.1% Triton X-100, 0.2 mM phenylmethylsulfonyl fluoride, pH 8.0). After sonication and centrifugation, the cell extract was collected. Subsequently, a 5 mL Ni–NTA column was equilibrated with binding buffer (50 mM Tris, 300 mM NaCl, pH 8.0) at a flow-rate of 5 mL•min$^{-1}$. The cell extract containing the target protein with His-tag was incubated with the equilibrated column packing for 1 h. Then, the products after incubation flowed through the column and were collected. The washing buffer was used to clean off unbound protein and the elution buffer was used to elute the target protein. After that, the purified components were dialyzed into a protein preservation buffer (50 mM Tris, 300 mM NaCl, pH 8.0). After dialysis, the protein was concentrated with polyethylene glycol 20,000, filtered using a 0.45-μm membrane, and stored at −80 °C.

## Debye shielding length ($\lambda_D$) calculation

$\lambda_D$ was calculated based on the formula[36]

$$\lambda_D = \sqrt{\varepsilon k_B T / q^2 c} \qquad (2)$$

where $\varepsilon$ is the electrolyte permittivity, $k_B$ is the Boltzmann constant, $T$ is the thermodynamic temperature, q is an elementary charge, and $c$ is the solution ionic strength.

## Real-time electrical measurements

The PNPase-modified SiNW FET device was covered by a polydimethylsiloxane cube, which contained a hole of about 2 mm in diameter as a reaction chamber[17,29]. The RNA substrates were kept warm at 65 °C and then quickly cooled to avoid the effect of the changeable structure. Then, a 60-μL RNA analog solution with a specific concentration was subsequently injected into the microchamber. The chamber temperature was precisely controlled with an INSTEC hot/cold chuck, which involved a proportion–integration–differentiation control system (±0.001 °C) and a liquid nitrogen cooling system. Using an HF2LI Lock-in Amplifier (Zurich Instruments), the source–drain voltage was kept at DC 0.3 V throughout all real-time electrical measurements. The current between the source electrode and the drain electrode was amplified through a DLL1211 preamplifier operating at $10^7$ V•A$^{-1}$ gain and recorded using the HF2LI Lock-in Amplifier equipped with a 5-kHz bandwidth low-pass filter at 28.8 or 7.2 kHz sampling rates.

## Statistical analysis

A low-pass Butterworth filter at 2 kHz was necessary to reduce the circuit signal noise of the raw data. The subsequent data processing was carried out using MATLAB 2016b. A QuB software was used to idealize the filtered data for obtaining the dwell time of each signal event and the number of total events based on the hidden Markov model. In detail, relying on the difference in the conductance of each state, each state was sorted out and fitted to the continuous transformation relationship. Based on the temporal resolution between two adjacent dots, the dwell times of each state were obtained. Finally, the values of the dwell time and current could be extracted as events. Origin 2019b was then used to import and analyze the extracted data. The data of the dwell time were sorted out into seven intervals and plotted in a histogram where the dwell time was in the x coordinate and the count was in the y coordinate. The statistical average dwell time was obtained by relying on the exponential decay function. All statistical data are presented as the mean ± SD.

## Angular velocity calculation

$$\omega = \arctan\left(\frac{y_2 - y_1}{x_2 - x_1}\right) \qquad (3)$$

$x_1$ and $y_1$ were coordinate values of the first data points; $x_2$ and $y_2$ were coordinate values of the second data points.

## Thermodynamic analysis

$\Delta H_T$ and $\Delta S_T$ were deduced based on the thermodynamic relationships:

$$-\Delta G_T / T = R\ln K_T = -\Delta H_T / T + \Delta S_T \qquad (4)$$

where $\Delta H_T$ is the change of enthalpy, equivalent to the negative value of the slope of the fitting curve; $\Delta S_T$ is the change of entropy, equivalent to the intercept of the fitting curve; $T$ is the thermodynamic temperature; $K_T$ is the transformation equilibrium constant from the RNA-free-PNPase structure to the PNPase RNA analog-bound structure, As to a first-order dynamic process, $K_T$ is the ratio of the concentration belonging to the two structures. The time interval, where data points emerged consecutively in one structure, was chosen as the instantaneous concentration, which is equal to the reciprocal of the occurrence number per unit of time (1/*p*). Therefore, $K_T$ is equal to the value of the ratio of 1/*p* (bound) to 1/*p* (free), where *p* is the occurrence proportion of each structure[17,29].

## Kinetic analysis

The activation energy $E_a$ was deduced by relying on the Arrhenius equation:

$$\ln(k) = \ln(A) - E_a/(RT) \tag{5}$$

where $A$ is a pre-exponential factor; $R$ is Avogadro's constant; $k$ is the hydrolysis reaction rate constant equal to the value of $1/\tau_I$ (hydrolysis reaction); and $T$ is the thermodynamic temperature. $E_a$ can be obtained from the slope of the linear fitting function to $\ln(k)$ and $1/RT$[17,29,50,56,57]. The KIE value was calculated based on the formulas:

$$KIE = k_H/k_D \tag{6}$$

where $k_H$, $k_D$ is respectively the hydrolysis reaction rate constant in the solvent of $H_2O$ or $D_2O$[46].

## Identification of the nucleotide

Totally, 150 couple values (dwell time and current) at every step were integrated to plot a fingerprint and 30 fingerprints were put together ordinally from the 5′ terminus of the sequence to obtain a collection. Relying on the contrast with the reference fingerprints obtained from the homogenous sequence, we could distinguish several nucleoside information in the sequence. Finally, in the same way, we could obtain ten collections from the bulk data. Through the information overlay of nucleoside sites from 10 collections, we could be competent to identify the information of 24 sites in the artificially designed sequence and 19 sites in a *mccA* gene sequence. All RNA analog sequences for experiments were provided by Synbio Technologies (Beijing) Co., Ltd.

## Statistics and reproducibility

Statistical analysis was conducted relying on the original data without randomization and blinding treatments. The experiments have separately proceeded in three independent devices to manifest the reproducibility.

## Reporting summary

Further information on research design is available in the Nature Portfolio Reporting Summary linked to this article.

## Data availability

All data supporting the findings of this study are available within the main manuscript and the supplementary files. Source data for the main manuscript figures are available as source data files. All source data (including for Supplementary Information figures) are available online from Zenodo at https://www.zenodo.org/record/7533205. Source data are provided with this paper.

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

## Acknowledgements

We thank Chao Guo, Ruoyao Xiong and Prof. Luhua Lai from Peking University for constructive discussions. We acknowledge primary financial supports from the National Key R&D Program of China (2021YFA1200101 and 2022YFE0128700) to X.G., the National Natural Science Foundation of China (22150013, 21727806, and 21933001) to X.G., the Tencent Foundation through the XPLORER PRIZE to X.G., the Natural Science Foundation of Beijing (2222009) and "Frontiers Science Center for New Organic Matter" at Nankai University (63181206) to X.G.

## Author contributions

X.G. and L.L. conceived and designed the experiments. Z.Y. W.L., L.Z., J.F., and D.Y. fabricated the devices and performed the device measurements. X.G., L.L., and Z.Y. analyzed the data and wrote the paper. All the authors discussed the results and commented on the paper.

## Competing interests

The authors declare no competing interests.
