## [Peer Review File · Nature Communications]

Revealing single-base degradation dynamics of RNA using single-exonuclease nanocircuits toward direct RNA sequencingREVIEWER COMMENTS

Reviewer #1 (Remarks to the Author):

In this manuscript, the authors reported the real-time monitoring of electrical signal readouts associated with mRNA analog binding/unbinding processes using single-PNPase attached SiNW-FET sensors. Using dwell-time and event probability analyses of bound and unbound states of PNPase and mRNA analog, the authors observed that the bound state becomes dominant and more stable with increasing temperature. In addition, the authors showed the strong dependence of the formation and stability of the PNPase-mRNA analog complex on pH range and the presence of metal ion, Mg²⁺. Moreover, the authors captured individual mRNA analog degradation events, distinguished different nucleosides, and identified the nucleoside sequence of a heterogeneous mRNA analog with ~ 80% accuracy. The electronic single-molecule sensing and single-base sequencing approach reported in this manuscript will be interesting to a broad readership. I have several comments and questions for the authors to consider in revising this manuscript.

1. It is not obvious how the two optical and electrical signals are perfectly synchronized (fig. 1f), even though the signal transduction mechanisms are completely different. What does the optical signal represent? There could be some time delay between two signals, different dwell duration, and/or some additional/missing signals between them. Please comment on this point.
2. Additionally, what is the electrical signal generation and transduction mechanism? Based on the text (Page 5), how do conformational changes affect the charge density variation? It would be nice to see a more detailed explanation of signal generation in terms of their p-type FET, protein charges, and their motions. It would be more convincing if the authors test several PNPase variants and identify the key residues, active sites, and their motions, which are responsible for the optical and electrical signal generation. Since the authors claimed SiNW-FET sensors are p-type, how the gate voltage was controlled during the experiment? It would be useful to see the IVg curves of the sensors.
3. Why do all electrical and optical signals show abrupt changes? The 3-dimensional conformational changes during binding, unbinding, and hydrolysis would be continuous motion, and thus electrical and optical signals are expected to be changed continuously like most single-molecule FRET signals.
4. The dimension and structure of the “suitable” nanogap are unclear (Page 5), which is very important for protein conjugation, protein conformational changes, and diffusion of ligand to it. If the nanogap is just the protein size (i.e., ~10 nm) and surrounded by 50 nm SiO₂ (or Su-8 or PMMA?), the diffusion of proteins to the gap with the correct orientation for the covalent bond to the SiNW would be difficult. In contrast, the gap is much larger than the protein size, it would be possible to be attached multiple proteins in the gap. How do the nonspecific adsorption of PNPase and mRNA analog on SiNW were controlled?
5. It is not obvious the activity buffer used in this experiment and its ionic strength due to the statement on page 5: “optimized through regulation of the solution ionic strength”. I assume that the authors lower the ionic strength of the buffer to increase the Debye length; however, what is the activity buffer used in the experiment? The authors need to show the activity assay data with the low ionic strength buffer. The blank buffer [20 mM Tris-HCl, 30 mM KCl] yields the Debye length of < 3 nm, not 11.3 nm, thus no charge motions or protein activity will be seen above 2.7 nm above the SiNW surface.

Reviewer #2 (Remarks to the Author):

There is considerable interest in the development and usage of technologies to trace biochemical processes at the single-molecule level. This is seen most clearly in the many adaptations of direct sequencing using nanopore arrays. Here the authors describe a new methodology to interrogate (sequence) mRNA based on the conjugation of a modified version of a single bacterial polynucleotide

phosphorylase (PNPase) to a semi-conductor sensor. 3' to 5' degradation of RNAs is recorded on a millisecond time scale and with single nucleotide resolution. This is a technically sophisticated study in terms of the construction and characterization of the devices. Beginning with simple homopolymers, the authors demonstrate differences in dwell times that can be exploited for sequencing. This is tested using a short 30 nucleotide RNA of a more complex sequence. The test mRNA analog is designed to omit secondary structures that may impede degradation. Considering that cellular mRNAs and ncRNAs often have extensive secondary structures this is a concern. I recognize that the sequencing represents proof-of-concept but as presented seems too artificial. Demonstrating the feasibility by using one or ideally more bona fide cellular RNAs would substantially enhance the impact of the manuscript.

RECOMMENDATIONS

The writing style is very convoluted, bordering on indecipherable at points. For instance, in the abstract, why say "transcribed information" when RNA would suffice? Likewise, jargon such as "terminal treatment of genetic information" or "ensemble average information", or "incorporation...into DNA polymerase I, and "systematic contrastive analyses" serves only to confuse and alienate the reader.

The potential use of these sensors as a new methodology for single-molecule RNA sequencing is very exciting however the feasibility is not adequately addressed. The test RNAs are only 30 nucleotides long and likely lack significant secondary structures. Given the fact that biologically relevant RNAs are often much longer and can have extensive secondary structure, the processivity needs to be addressed.

There is no discussion of how RNAs are targeted to the tethered PNPase. The title and abstract specifically refer to mRNAs, which implies there is some selectivity. A potential limitation to the approach would be if there is bias in the RNA 3'-ends that most efficiently engage with the exonuclease.

Reviewer #3 (Remarks to the Author):

The authors present an application of a SiNW FET detector coupled to the 3' -> 5' exonuclease PNPase. PNPase is an evolutionary conserved nuclease that has been studied in several organisms over the past decades, but high-resolution RNA decay dynamics are still hard to come by and studying PNPase in this regard is a reasonable start. Furthermore, developing new direct RNA sequencing strategies possibly expanding the toolset for RNA analysis sounds appealing. Hence, the topic of this manuscript is interesting in general, but the presentation of the data in figures, methods and main text is of low quality that I cannot recommend this manuscript for publication. In addition, the mechanistic conclusions are largely based on strong interpretations and speculation without experimental follow-up to support them.

Specific comments:

For me as a molecular biologist I could not develop a sense for the scalability and reproducibility of the method from the manuscript:

How many devices were made? How many different PNPase molecules were investigated? Are dwell time analyses based on readings from a single PNPase over time or were multiple readings combined from the same PNPase molecule or multiple different ones?

I read the text as if everything comes from several binding/unbinding or RNA decay reactions on one nanowire, i.e. a single PNPase!

I appreciate the characterization of conductance properties in different conditions, e.g. salt

concentration, pH and temperature. The results are primarily presented as histograms of dwell times with the average dwell time given in each panel in the supplement. The high number of plots makes it almost impossible to extract the information the authors are talking about in the main text. I suggest plotting the various conditions on the x-axis, e.g. 5 different temperatures, and average dwell times with error bars on the y-axis. This yields a scatter plot with a clear visualization of dwell time dependence on the tested parameter. This would allow to see the increase or decrease the authors are talking about in the text. Then the authors could add some of these aspects into the main part of the manuscript to improve readability.

Confusingly, the authors present histogram data similar to the supplement in the main figure as dots (compare 2a, 3c with S6, S8, S10 etc), that is inconsistent and not described accurately in the legend. These plots are very confusing, e.g. where is 0 – to the left or to the right?

Furthermore, the authors try to have a color scheme based on low/intermediate and high conductance scheme, but switch it in basically every figure, which makes no sense and is confusing. Examples for this are Fig 1c, 2a, 2b and the dwell time distributions in the supplement.

Often the axes in the plots are missing their correct labels. Sometimes only units are given (e.g. 2b left, 2c, 3c bottom left, 3d, 3f), 4b heatmap is missing everything. Intensity scale in heatmap and density map in 4b is not labeled. S22 top panel x-axis label missing. Figure 3d has unlabeled graphs & axis that prevent following the main text section p9, lines 9 onwards.

Figure 3f & 3g, 4e & 4f switched in text, respectively

Figure 1c – immediate = intermediate?

Selecting degradation processes that involve 30 steps for a 30nt RNA for dwell time analysis seems reasonable as a first step, but how rare/frequent were these compared to less processive events? Understanding what is due to technical ‘artefacts’ or real biology might be interesting.

Main text writing – in general, the authors jump quickly from one thought / tested condition to the next without explaining their logic, defining hypotheses, or summarizing conclusions. I give a few examples here:

- For example, page 7 line 17-22 is very unclear to me as a non-expert, not explained in the methods and I cannot see what the motivation is
- Repeatedly dwell time changes are interpreted on a mechanistic basis with specific molecular interactions being affected. This is especially apparent on page 13, line 3-16. The authors correctly state that they “speculate” there, but this speculation then leads to the abstract with the words “a binding event [...] was discovered for the first time”. There is no further experimental or analytical follow-up that justifies the claim in the abstract.
- Page 9, line 19 “show” wrongly used – maybe “could indicate” ok
To use “show” further evidence is needed, e.g. by modifying the potential binding sites genetically and testing the effect in this single molecule assay
- Page 13 line 20 citation is missing, and why was this analysis done in this context not given
- Thermodynamic and kinetic analysis are not comprehensive to a non-expert
- Unclear what are 150 or 20 sample volumes – replicates, same PNPase and 20 RNAs...?
- The last results section on the feasibility of the SiNW device for sequencing is very unclear to me, nor do I understand what 5d and 5e show as results
- From what I understand from this section, several rounds of analysis of the same sequence are needed to identify the nucleotide sequence reliably – how would this be compatible with sequencing real transcripts that are present in few copy numbers without amplification and in a mixed pool?

Methods are too brief to allow for reproduction, e.g. it would be good to have a precise explanation how dwell times are calculated, as these form the basis for many aspects of the manuscript.

Listed below are the major changes in the new manuscript:

1. We have carefully complemented additional measurements and confirmed mechanistic hypotheses, relying on the single-molecule modification of a mutant PNPase, where R93 and R97 were mutated to G93 and G97.

Please see Page 17 in the revised main text and Supplementary Fig. 28.

2. We also complemented electrical measurements for a synthesized cellular RNA (i.e. *mccA* gene), which yielded a ~79.17 % identified efficiency.

Please see Page 19 in the revised main text and Supplementary Figs. 41–51.

3. We have supplied the electrical measurement results for the mixed sample containing an artificially designed sequence and a *mccA* gene sequence, demonstrating the single-molecule precision in our work.

Please see Pages 19–20 in the revised main text and Supplementary Figs. 52–53.

4. We complemented the discussion about the step size of the degradation process to better analyze the real biological behavior.

Please see Page 13 in the revised main text and Supplementary Fig. 26 in the revised Supplementary Information.

5. We have improved the detailed description and graphs about the fabrication and modification process of the sensors for a better acknowledgement of our work.

Please see Page 5 in the revised main text and Supplementary Figs. 2 and 4.

6. We have added a detailed description of the synchronized recording process of optical and electrical signals.

Please see Page S7 in the revised Supplementary Information.

7. The detailed analysis routine from the electrical testing data to the calculation of the dwell time has been supplied in the method of the main text.

Please see Pages 23–25 in the revised main text.

8. The sequencing analysis routine of the electrical testing data has also been carefully described in the method of the main text.

Please see Page 25 in the revised main text.

9. We have supplied the table of assay data about the regulation of the Debye length in the Supplementary Information and improved the description of the regulation method.

Please see Pages 6 in the revised main text and Supplementary Stable 2 in the revised Supplementary Information.

10. The identification graph for the sequence has been changed to the seq-logo graph for a better illustration of the identification result.

Please see Fig. 5e in the revised main text.

11. We unified the color scheme based on low (yellow) / intermediate (green) and high conductance (purple) and extended these to all associated figures.

Please see Figs. 2–3 in the revised main text and Supplementary Figs. 7, 8, 10, 12, and 21–25 in the revised Supplementary Information.

12. We have plotted a scatter diagram of the average dwell times attached with error bars for the binding process and degradation process under different conditions.

Please see Supplementary Figs. 5 and 19 in the revised Supplementary Information.

Response letter

General Reply:

We sincerely thank all the reviewers very much for their precious time involved in reviewing the manuscript and the valuable feedbacks that we have used to improve the quality of our manuscript. The reviewers' comments are laid out below in black font and specific concerns have been numbered. Our responses are given in blue font and changes/additions to the manuscript are given in the italic text. The places of changes/additions for the manuscript are guided with bold font in the response letter and highlighted in the revised version. Modified graphs were marked with the highlighted legend in the main text and Supplementary information. We hope they will be satisfied with this revised version.

Reviewer #1 (Remarks to the Author):

Comments:

“In this manuscript, the authors reported the real-time monitoring of electrical signal readouts associated with mRNA analog binding/unbinding processes using single-PNPase attached SiNW-FET sensors. Using dwell-time and event probability analyses of bound and unbound states of PNPase and mRNA analog, the authors observed that the bound state becomes dominant and more stable with increasing temperature. In addition, the authors showed the strong dependence of the formation and stability of the PNPase-mRNA analog complex on pH range and the presence of metal ion, Mg^{2+} . Moreover, the authors captured individual mRNA analog degradation events, distinguished different nucleosides, and identified the nucleoside sequence of a heterogeneous mRNA analog with $\sim 80\%$ accuracy. The electronic single-molecule sensing and single-base sequencing approach reported in this manuscript will be interesting to a broad readership. I have several comments and questions for the authors to consider in revising this manuscript.”

Response: Thank this reviewer very much for his/her time involved in reviewing the manuscript and the very encouraging comments on the merits. The professional suggestions were very precious to us in improving the accessibility of our manuscript. We also appreciate his/her clear and detailed feedbacks and hope that the explanations have fully addressed all of his/her concerns.

In the remainder of this section, we will discuss each of the comments along with our corresponding responses. The relevant revised contents in the main text are provided below in italic font for the quick reference.

1. It is not obvious how the two optical and electrical signals are perfectly synchronized (fig. 1f), even though the signal transduction mechanisms are completely different. What does the optical signal represent? There could be some time delay between two signals, different dwell duration, and/or some additional/missing signals between them. Please comment on this point.

Response: Thanks a lot for the detailed review and useful suggestions for our article. We apologize for the incomplete description in the former version and have carefully revised the description of the synchronized recording process of both optical and electrical signals in the section of “Optical characterization” **in the Supplementary Information on Page 7.**

In detail: “*Subsequently, the device was placed on the microscope objective stage. The target electrode pair were caught by two extension probes, which were appended to the signal source and*

drain sockets of the UHFLI lock-in amplifier, respectively. A Nikon Ni-E microscope with a $\times 100$ objective lens was positioned in close contact with the coverslip on the device by feat of the lens oil. The signal collection area of the stochastic optical reconstruction microscopy was zoomed on the modified portion of the single silicon wire. Then, a 548 nm laser was employed to excite the device and an EMCCD (Andor) was used to receive the feedback emitted light and the fluorescence spectra with a 50 ms exposure resolution. The process control cable of a stochastic optical reconstruction microscopy was connected to the trigger terminal of the UHFLI lock-in amplifier (high-speed sampling rate: $57600 \text{ Sa}\cdot\text{s}^{-1}$) to provide synchronous triggering. Once the trigger was switched, the fluorescent and electronic signals of the target location were simultaneously recorded within 5 minutes. After that, through the reconstruction and analysis of the optical pictures using the Advanced Research software, a single fluorescence spot and an enlarged image without the background were merged.”

So, under the illumination of a 548 nm laser, the optical signals above the single PNPase modified portion represented the residence events of a single RNA analogue labelled with 5'-tetramethylrhodamine azide (excitation wavelength 548 nm) inside the PNPase protein, which is consistent with the cause of electrical signal generation. As commented by this reviewer, some time delay between two signals indeed exists in our measurement processes because of the temporal resolution discrepancy (**shown in Figure R1 middle**). In addition, some transient processes such as Mode 1 or 3 were unable to be caught by the optical recording technique due to the limit of the temporal resolution. Therefore, the synchronization of the optical and electrical signals can be used to certify that the electrical signals originate from the bind action of RNA analogue with PNPase at a single-protein level and achieve more convincing data. Because of a higher temporal resolution of electrical measurements, we prefer to use electrical signals for statistical analyses in the work.

Figure R1. The partially enlarged detail (0.5 s) of Figure 1f in the main text (middle panel) and further enlargement parts for the corresponding changes in the middle panel (left panel and right panel).

2. Additionally, what is the electrical signal generation and transduction mechanism? Based on the text (Page 5), how do conformational changes affect the charge density variation? It would be nice to see a more detailed explanation of signal generation in terms of their *p*-type FET, protein charges, and their motions. It would be more convincing if the authors test several PNPase variants and identify the key residues, active sites, and their motions, which are responsible for the optical and electrical signal generation. Since the authors claimed SiNW-FET sensors are *p*-type, how the gate voltage was controlled during the experiment? It would be useful to see the IV_g curves of the sensors.

Response: Thank this reviewer once again for his/her good suggestions on improving the accessibility of our manuscript. In terms of our *p*-type FET, we have improved our description in

the main text in Lines 7–10 of Page 7 as follows “*In terms of the p-type FET, this structural change can enhance the surface electronegativity of PNPase (Fig. 1b), which functions as an additional gate and thus leads to the observed elevation in the device conductance (Figs. 1c and 2a).*” and **in Lines 11–13 of Page 11** as follows “*This ratchet movement of RNA might indirectly cause the enhancement of the surface electronegativity of PNPase, which induces a high conductance state.*” During the electrical measurement process, we only relied on the intrinsic charge change of PNPase to adjust the conductance of the device without applying an additional gate voltage. The $I-V_g$ curves of the device were also attached **in Figure S3 in the Supplementary Information.**

In the revised version, we complemented the electrical measurement results of the mutant PNPase, where R93 and R97 were mutated to G93 and G97, certifying that the 3' terminus coordination action of the RNA was indeed generated in the active site of the PNPase. In addition, the mutation of R93 and R97 also impairs the sustainability of the single base degradation behavior. The measurement data and discussion were supplied **in the Supplementary Fig. 28 and the main text in Lines 3–20 of Page 17**, respectively.

“In order to investigate whether R93 and R97 amino acids participate in the coordination with nucleosides, we mutated R93 and R97 into G93 and G97, which do not have the structure to form hydrogen bonds. The real-time degradation process of the mutant PNPase for poly(A)₃₀ was monitored under the same experimental condition (see Supplementary Fig. 28a). Through the fitting for the intermediate state and high state, we found that both states presented a single exponential decay process, which indicated that there was a process lost after mutation (see Supplementary Fig. 28b). The dwell time of the intermediate state ($\tau = 78.84 \pm 4.17$ ms) was augmented in contrast to the value ($\tau_1 = 46.33 \pm 3.19$ ms, $\tau_2 = 8.86 \pm 0.25$ ms) before mutation, showing that the degradation process was delayed after mutation. The reduction of the high-state proportion and the shortening of the step length demonstrated that the stability of single base degradation was also affected (see Supplementary Fig. 28b). After the analysis of the electrical measurement data for the four types of the homogenous sequence, the fingerprints of the intermediate state were presented in Supplementary Fig. 28c. There was not an obvious difference in the fingerprints of the four types of homogenous sequences. In general, these results manifested that R93 and R97 participated in the coordination with the nucleosides of the RNA analog and played a key role in maintaining the stability of a single base degradation.”

In this work, we focus on the analysis of thermodynamic and kinetic actions of PNPase and RNA analog at the single-molecule level and then try our best to realize the distinguishment at single-base resolution. We trust it is glamorous to explore some key activity sites and residues inside the enzyme for the identification of respective regional functions during the enzymatic reaction process at a single protein level. We believe there will be good opportunities to achieve the breakthroughs, relying on genetic engineering techniques to mutate key sites and precise analysis by feat of more advanced algorithmic systems in the future. We will continue our effort in this direction.

3. Why do all electrical and optical signals show abrupt changes? The 3-dimensional conformational changes during binding, unbinding, and hydrolysis would be continuous motion, and thus electrical and optical signals are expected to be changed continuously like most single-molecule FRET signals.

Response: Thank this reviewer for the meticulous consideration of this point. We agree with this reviewer's opinion about the fact that the conformational changes of the protein are continuous

motions. Aiming to show this changing process, we have highlighted electrical transform data of the RNA binding and releasing processes in Figure 2c in the main text. Relying on trajectory clustering algorithms, we obtained the full widths at half maxima during the RNA analog binding process (~1.79 ms) and RNA analog releasing process (~2.13 ms). The electrical conversion processes of Figure 1f were indeed continuous, but they were compressed to be undiscernible because of the huge data volume. In addition, the conformational changes in the binding and releasing processes are merely associated with bulk expansion and restoration, which show a lower entropy increment mode in comparison with the structure changes involving the unfolding and folding actions of the polypeptide chains. Therefore, the conversion changes in the current value are shown to be faster. In order to better clarify this question, we amplify the simultaneously recorded fluorescent and electrical signals in the following figure (Figure R1) from Figure 1f in the main text, clearly showing the gradual conformational changes.

Figure R1. The partially enlarged scatter detail (0.5 s) of figure 1f in the main text (middle panel). Enlargement panels for the middle panel correspond to the conformational changes (left panel and right panel).

4. The dimension and structure of the “suitable” nanogap are unclear (Page 5), which is very important for protein conjugation, protein conformational changes, and diffusion of ligand to it. If the nanogap is just the protein size (i.e., ~10 nm) and surrounded by 50 nm SiO₂ (or Su-8 or PMMA?), the diffusion of proteins to the gap with the correct orientation for the covalent bond to the SiNW would be difficult. In contrast, the gap is much larger than the protein size, it would be possible to be attached multiple proteins in the gap. How do the nonspecific adsorption of PNPase and mRNA analog on SiNW were controlled?

Response: Thank this reviewer very much for the precious suggestion on improving the accessibility of our work. For a better illumination of a single PNPase modification method, we have revised some important descriptions **on Lines 7–16, Page 5 in the main text** and the partial descriptions on the device fabrication as well as the immobilization process of a single *E. coli* PNPase protein **on Pages 3 and 4 in the Supplementary Information**.

In detail: “Through precise high-resolution electron-beam lithography and wet etching, a nanoscale special area terminated with Si–H and suitable in size for the conjunction with a single protein molecule was generated on the core-shell SiNWs. After removing the polymethyl methacrylate layer with vast acetone, this special surface of SiNWs was successively modified by undecylenic acid hydrosilylation, N-hydroxysuccinimide esterification, and maleimide immobilization. Ultimately, by feat of the confinement effect, a single PNPase molecule was conjugated to the molecular bridge on the surface of the SiNW through a thiol–maleimide–Michael addition, relying on a mutated cysteine residue at the bottom of the α -helical domain that has no effect on the activity of the enzyme.”

“Considering the testing condition that the drain current was measured in the buffer solution, a negative resist (SU-8, 2002) was utilized to open a window by photolithography to expose the intact surfaces of silicon wires for following modifications and protect the majority of the surface to prevent a generation of the electrical leakage. The windows between electrodes also functioned as a reactive chamber for the measurement of the enzymatic reaction process *in vitro*.”

We also supplied the detailed fabrication process of the device (Figure R2) and the locating process of the confinement area (Figure R3) in **Supplementary Figs. S2 and S4**.

In detail:

Figure R2. The brief process of mechano-sliding transfer of SiNWs and FET array fabrication.

Figure R3. Schematic demonstration of the strategy used for surface functionalization and single PNPase protein immobilization.

We listed a three-part explanation for a better response to the questions:

I. To passivate the contact interface of the gold electrodes, a 50 nm thick SiO₂ protective layer was only deposited on the surface of the gold electrodes. The deposition layers (SiO₂, Au, Cr layers) between the electrodes and above the SiNWs were removed by cleaning away the photoresist (Figure R2).

II. Considering the testing condition that the drain current was measured in the buffer solution, to protect the majority of the surface of the device and prevent a generation of the electrical leakage,

a negative resist (SU-8) was utilized to open a window only between the electrodes by photolithography to expose the intact surfaces of silicon wires for following modifications of the SiNWs (Figure R2). The windows above the SiNWs also functioned as a reactive chamber for the measurement of the enzymatic reaction process *in vitro*.

III. Through the precise high-resolution electron-beam lithography (EBL) and full development in a mixture of water/isopropanol, a certain nanogap (10–20 nm) was generated in the PMMA layer on the surface of the SiNWs. Then, the surface of the SiNWs inside the nanogap was wet etched in a buffered HF solution for about 7 s to remove the amorphous SiO₂ shell and expose the silicon surface (10–20 nm) that is terminated with Si–H bonds (The optimization of the etched strategy to obtain a 10–20 nm scale surface had been reported in our previous work: “Point decoration of silicon nanowires: An approach toward single-molecule electrical detection”, *Angew. Chem. Int. Ed.* **2014**, *53*, 5038). Finally, the whole PMMA layer was cleaned away with acetone and then only a special surface (10–20 nm) suitable for the confinement conjunction with a single PNPase molecule in size was left on the surface of the SiNW.

In summary, through the EBL and etching, we can obtain a special area on the surface of SiNWs for the following confinement conjunction. A 50 nm SiO₂ layer was deposited only on the surface of metal electrodes. The PMMA layer above the SiNWs was cleaned away by acetone in the end. The Su-8 covered the majority of the device, but a window was opened above the SiNWs. In addition, the redundant PNPase can be cleaned away by washing with buffer solution as well as a weak ultrasonic treatment after the sufficient conjunction reaction. During the measurement process, the surface of the SiNWs showed electronegativity originating from the existence of the vast hydroxy, which well prevents the nonspecific adsorption of the electronegative RNA substrate by feat of the electrostatic repulsion.

5. It is not obvious the activity buffer used in this experiment and its ionic strength due to the statement on page 5: “optimized through regulation of the solution ionic strength”. I assume that the authors lower the ionic strength of the buffer to increase the Debye length; however, what is the activity buffer used in the experiment? The authors need to show the activity assay data with the low ionic strength buffer. The blank buffer [20 mM Tris-HCl, 30 mM KCl] yields the Debye length of < 3 nm, not 11.3 nm, thus no charge motions or protein activity will be seen above 2.7 nm above the SiNW surface.

Response: Thanks a lot for the constructive advice on the advancement of our manuscript. We apologize for our insufficient illumination of the optimized data ascribed to the regulation of the solution ionic strength. We chose the Tris-HCl buffer because of the lower ionic strength and the adjustment of the concentration allows us to realize the increment of the Debye length. We have supplied the table of the assay data (**Table R1**) about the regulation of the Debye length **in the Supplementary Information (Table S2)** and improved the description of the regulation method **in Lines 1–4, Page 6 in the main text.**

Table R2. A table of the assay data about the regulation of the Debye length.

Tris-HCl pH=7.6	λ_D (nm)	ionic strength (mM)
×1	1.13	59.30
×0.1	3.58	5.93
×0.01	11.32	0.59

In detail: *“(Table S2). As is evident, the Debye length increased with the decrement of the ionic strength. Through the dilution of the Tris-buffer (20 mM Tris–HCl, pH 7.6, 30 mM KCl) by 100-fold, the Debye shielding length could be adjusted to about 11.3 nm.”*

We would like to take this opportunity to thank this reviewer very much for his/her precious time and for the great opportunity for us to improve the manuscript. We hope this reviewer will find this revised version satisfactory.

Sincerely,

The Authors

-----End of Reply to Reviewer #1-----

Reviewer #2 (Remarks to the Author):

There is considerable interest in the development and usage of technologies to trace biochemical processes at the single-molecule level. This is seen most clearly in the many adaptations of direct sequencing using nanopore arrays. Here the authors describe a new methodology to interrogate (sequence) mRNA based on the conjugation of a modified version of a single bacterial polynucleotide phosphorylase (PNPase) to a semi-conductor sensor. 3' to 5' degradation of RNAs is recorded on a millisecond time scale and with single nucleotide resolution. This is a technically sophisticated study in terms of the construction and characterization of the devices. Beginning with simple homopolymers, the authors demonstrate differences in dwell times that can be exploited for sequencing. This is tested using a short 30 nucleotide RNA of a more complex sequence. The test mRNA analog is designed to omit secondary structures that may impede degradation. Considering that cellular mRNAs and ncRNAs often have extensive secondary structures this is a concern. I recognize that the sequencing represents proof-of-concept but as presented seems too artificial. Demonstrating the feasibility by using one or ideally more bona fide cellular RNAs would substantially enhance the impact of the manuscript.

Response: Thank this reviewer very much for his/her precious time involved in reviewing the manuscript and very encouraging comments on the merits. Your professional suggestions were very precious to us in improving the accessibility of our manuscript. We also appreciate his/her clear and detailed feedbacks and hope that the explanations have fully addressed all of his/her concerns.

In the remainder of this section, we will discuss each of the comments along with our corresponding responses and the relevant revised contents in the main text are provided below in italic font for the quick reference.

RECOMMENDATIONS

1. The writing style is very convoluted, bordering on indecipherable at points. For instance, in the abstract, why say “transcribed information” when RNA would suffice? Likewise, jargon such as “terminal treatment of genetic information” or “ensemble average information”, or “incorporation...into DNA polymerase I, and “systematic contrastive analyses” serves only to confuse and alienate the reader.

Response: Thank this reviewer very much for the detailed review and the important suggestions for improving the readability of our manuscript. We sincerely apologize for the inaccurate expressions in our manuscript. We have carefully and thoroughly proofread the manuscript to correct all the inaccurate expressions. We hope these revisions would help readers to get a better understanding of our work.

In detail:

(1). *“Terminal processing of RNA is decisive in guaranteeing high-fidelity translation of genetic information in living organisms.”* **in Lines 2–3 on Page 2 in the main text.**

(2). *“Comprehensive deciphering of the mechanism of RNA degradation is critical for research on the details of the high-fidelity principle for transcriptome information transmission and precise regulation routine of the physiological environment;”* **in Lines 6–8 on Page 3 in the main text.**

(3). *“However, traditional macroscopic technologies used to analyze the conformational changes during enzyme–substrate catalytic reactions usually produce collective information about the samples, which can disguise the detailed properties of individual enzymes.”* **in Lines 12–15 on**

Page 3 in the main text.

(4). “the binding mechanism of the DNA polymerase I with deoxyribonucleoside triphosphate analogs.” **in Lines 2–3 on Page 4 in the main text.**

(5). “CGAUCUUCAUUGCCAAGCGGCUAGCUCAAA-3’ sequence could be identified—indeed, through ordinal contrast of these fingerprint profiles, 24 nucleosides could be identified effectively.” **in Lines 7–9 on Page 19 in the main text.**

2. The potential use of these sensors as a new methodology for single-molecule RNA sequencing is very exciting however the feasibility is not adequately addressed. The test RNAs are only 30 nucleotides long and likely lack significant secondary structures. Given the fact that biologically relevant RNAs are often much longer and can have extensive secondary structure, the processivity needs to be addressed.

Response: Thanks a lot for the constructive suggestions. These precious suggestions give us a lot of inspiration. The high-throughput de novo sequencing is our original intention and persistent goal in our subsequent studies. We hope to eventually achieve it by relying on the upgrade of the device sensitivity and the promotion of the data analysis flux in the future.

In order to demonstrate the feasibility of sequencing RNAs with the significant secondary structure at the single-molecule resolution, we have synthesized a cellular sequence (i.e. *mccA* gene), which is responsible for encoding an oligopeptide of seven amino acids corresponding to the primary structure of microcin C7 in *Escherichia coli* cells (Ref. *González-Pastor, J., San Millán, J. & Moreno, F. Nature 369, 281 (1994)*). The potential secondary structures were shown **in Supplementary Fig. S40**. After electrical measurements and data analysis, the sequence analysis collections were shown **in Supplementary Figs. S41–S51**. The identified efficiency of the *mccA* gene sequence yields about 79.17 %.

We also supplied the electrical measurement results, which originated from the mixed samples involving an artificially designed sequence (30 bp in length) and a *mccA* gene sequence (24 bp in length). The complete single base degradation processes (between two adjacent low states) were mainly presented with thirty steps (marked with blue) and twenty-four steps (marked with brown) shown **in Supplementary Fig. S52**. The statistical distributions of the sequence length were shown as located around sites at 30 bp and 24 bp in length (**Supplementary Fig. S53**). These results demonstrated the single-molecule precision in our study again. We added relevant discussions **in Lines 11–22 on Page 19 and in Lines 1–8 on Page 20 in the main text.**

In detail: “In addition, for a more realistic sequencing for transcriptome information relying on this technique, we synthesized a cellular RNA sequence 5’-UACGCAUGACCAUUA CGUUUGAUU-3’ (i.e. *mccA* gene), which is responsible for encoding an oligopeptide of seven amino acids corresponding to the primary structure of microcin C7 in *Escherichia coli* cells.^{54,55} The potential secondary structures were shown in Supplementary Fig. 40. The RNA substrates were kept warm at 65 °C and then quickly cooled to obtain a few unfold sequences to avoid the repellency effect of the PNPase on the changeable secondary RNA.⁵⁶ After electrical measurements and data analysis, the sequence analysis collections were shown in Supplementary Figs. 41–50. Finally, the identified efficiency of the *mccA* gene sequence could reach about 79.17 % (Fig. 5e and Supplementary Fig. 51). Through the electrical measurement for the mixed sample involving an artificially designed sequence (30 bp in length) and a *mccA* gene sequence (24 bp in length), we found that the complete single base degradation processes were mainly characterized by thirty steps (marked with blue) and

twenty-four steps (marked with brown) as shown in Supplement Fig. 52. The statistical distributions of the step length were also shown as located around sites at 30 bp and 24 bp in length (Supplementary Fig. 53). These results demonstrated the single-molecule precision of this electrical testing technique. Collectively, these results demonstrate the promising capability of single-exonuclease nanocircuits in distinguishing different nucleosides at single-base resolution.”

3. There is no discussion of how RNAs are targeted to the tethered PNPase. The title and abstract specifically refer to mRNAs, which implies there is some selectivity. A potential limitation to the approach would be if there is bias in the RNA 3'-ends that most efficiently engage with the exonuclease.

Response: Thanks a lot for these precious comments. After our deliberative consideration of the manuscript, we substituted “mRNA” by “RNA” in our work to improve the precision. Because the PNPase could decay small fragment RNA without special selectivity for functional categories of RNA in biological processes. As mentioned by this reviewer, during our electrical testing process, some degradation processes with less than 30 steps were observed more than once, which were associated with a certain bias generation. Therefore, we focused on the statistical analysis of the processes, which had a precise 30-step degradation process to improve the accuracy of the results. During the measurement process, the detailed targeting behaviors were not observed, which is possible because the effect of the targeting recognition behaviors on the structure change of the PNPase was too weak to be detected. In the following work, we would divide the PNPase into different domains to try to reveal the targeting recognition mechanism of RNAs.

We would like to take this opportunity to thank this reviewer very much for all the time involved and for this great opportunity for us to improve the manuscript. We hope this reviewer will find this revised version satisfactory.

Sincerely,

The Authors

-----End of Reply to Reviewer #2-----

Reviewer #3 (Remarks to the Author):

The authors present an application of a SiNW FET detector coupled to the 3' -> 5' exonuclease PNPase. PNPase is an evolutionary conserved nuclease that has been studied in several organisms over the past decades, but high-resolution RNA decay dynamics are still hard to come by and studying PNPase in this regard is a reasonable start. Furthermore, developing new direct RNA sequencing strategies possibly expanding the toolset for RNA analysis sounds appealing. Hence, the topic of this manuscript is interesting in general, but the presentation of the data in figures, methods and main text is of low quality that I cannot recommend this manuscript for publication. In addition, the mechanistic conclusions are largely based on strong interpretations and speculation without experimental follow-up to support them.

Response: Thank this reviewer very much for his/her precious time involved in reviewing the manuscript and very encouraging comments on the merits. The professional suggestions were very precious to us in improving the accessibility of our manuscript. We also appreciate his/her clear and detailed feedbacks and hope that the explanations have fully addressed all of his/her concerns.

In the remainder of this section, we will discuss each of the comments along with our corresponding responses and the relevant revised contents in the main text are provided below in italic font for the quick reference.

Specific comments:

1. For me as a molecular biologist I could not develop a sense for the scalability and reproducibility of the method from the manuscript: How many devices were made? How many different PNPase molecules were investigated? Are dwell time analyses based on readings from a single PNPase over time or were multiple readings combined from the same PNPase molecule or multiple different ones? I read the text as if everything comes from several binding/unbinding or RNA decay reactions on one nanowire, i.e. a single PNPase!

Response: Thanks a lot for the useful comments. We feel really sorry for our incomplete descriptions. We have added the description of the obtained results **in the legends of Figs. 2 and 3 in the main text and Supplementary Figs. 5, 6, 19 and 20 as follows:** *“mean of n = 3 technical replicates from three different single-PNPase-modified SiNW devices, error bars indicate s.d.”*. Our analyzed results, such as dwell time, proportion, entropy change and enthalpy change were deduced by electrical measurement data from three different devices (i.e. three different PNPase molecules) in order to manifest the reproducibility of results. The discrepancies among different measurements were reflected by the error bars in the corresponding figures.

2. I appreciate the characterization of conductance properties in different conditions, e.g. salt concentration, pH and temperature. The results are primarily presented as histograms of dwell times with the average dwell time given in each panel in the supplement. The high number of plots makes it almost impossible to extract the information the authors are talking about in the main text. I suggest plotting the various conditions on the x-axis, e.g. 5 different temperatures, and average dwell times with error bars on the y-axis. This yields a scatter plot with a clear visualization of dwell time dependence on the tested parameter. This would allow to see the increase or decrease the authors are talking about in the text. Then the authors could add some of these aspects into the main part of

the manuscript to improve readability. Confusingly, the authors present histogram data similar to the supplement in the main figure as dots (compare 2a, 3c with S6, S8, S10 etc), that is inconsistent and not described accurately in the legend. These plots are very confusing, e.g. where is 0 – to the left or to the right? Furthermore, the authors try to have a color scheme based on low/intermediate and high conductance scheme, but switch it in basically every figure, which makes no sense and is confusing. Examples for this are Fig 1c, 2a, 2b and the dwell time distributions in the supplement.

Response: We sincerely thank this reviewer very much for the valuable suggestions to improve the quality of our manuscript. We totally approve all the suggestions and have plotted the average dwell times attached with error bars of the binding process and degradation process under different conditions in **Supplementary Figs. 5, 6, 19, and 20**. In addition, we have added these into the main part of the manuscript in the corresponding discussion positions in **Line 21 on Page 7, Lines 11 and 22 on Page 9, Line 20 on Page 10, Line 16 on page 12, Line 1 on Page 13, and Line 19 on Page 13**. In addition, to improve the accuracy of the data representation, we complemented the histogram data of the temperature gradient experiments under the binding and degradation processes in **Supplementary Figs. 7 and 21**. We have carefully revised the plotting of the figures and improved illuminations about the origin of the coordinates, which were exhibited in **the legends of Figs. 2 and 3**. Finally, we have uniformed the color scheme based on low (yellow)/intermediate (green) and high conductance (purple) and extended these to all associated figures (**Figs. 1, 2, and 3 in the main text as well as Supplementary Figs. 7, 8, 10, 12, and 21–25**) for a better expression of the results. Thank you very much again.

3. Often the axes in the plots are missing their correct labels. Sometimes only units are given (e.g. 2b left, 2c, 3c bottom left, 3d, 3f), 4b heatmap is missing everything. Intensity scale in heatmap and density map in 4b is not labeled. S22 top panel x-axis label missing. Figure 3d has unlabeled graphs & axis that prevent following the main text section p9, lines 9 onwards. Figure 3f & 3g, 4e & 4f switched in text, respectively. Figure 1c – immediate = intermediate?

Response: Thanks a lot for the careful reading and constructive comments. We sincerely apologize for our carelessness in plotting graphs. We have complemented the labels in all figures, such as **Figs. 1, 2b, 2c, 2d, 3c, 3d, 3f, 4b (coordinates and intensity scale), 5d and Supplementary Figs. 14, and 29–39**. The abscissa axis represents the temperature and the vertical axis represents the dwell time in Figs. 2b left and 3c bottom left. The dots of the dwell times were plotted as a jitter distribution to avoid the overlay of the data. We improved the graphs and added labels to them. Supplementary Fig. 22 was also revised in the same way. “immediate” has been changed to “intermediate” in Figure 1c. Based on the progress of the discussion, the discussions about Figs. “3g, 3f” were switched in **Lines 8–22 on Page 14 in the main text** (Figures 4e and 4f were not involved in the marks of the figure, so we have not revised them). We hope these improvements can help readers to understand the content of the all graphs. Thank this reviewer once again in earnest.

4. Selecting degradation processes that involve 30 steps for a 30nt RNA for dwell time analysis seems reasonable as a first step, but how rare/frequent were these compared to less processive events? Understanding what is due to technical ‘artefacts’ or real biology might be interesting.

Response: Thank this reviewer for the valuable suggestions. We supplemented the step statistics of the degradation process under different conditions in **Supplementary Fig. 26** and complemented the discussion about the impact of the different conditions on the single-base degradation process

in Lines 7–16 on Page 13 in the main text.

In detail: *“Through the counting of the step length in the complete degradation processes (between two adjacent low states) in 20 seconds, we found that the persistence of single base degradation could be improved by the optimization of testing conditions (Supplementary Fig. 26). Under the weak acidic conditions, the events of the complete degradation process were more frequent, but the degradation products were dominated by small fragments (less than 30 steps). These indicated the whole degradation process of RNA analog was faster under weakly acidic conditions, but the sustainability as a single base step was poor. Unlike pH conditions, the improvement in the temperature did not significantly impact the frequency of the complete degradation process, but mainly contributed to the persistence of the single base degradation.”*

Relying on the electrical measurement of the mutant PNPase (R93, R97 to G93, G97), the steps of the single base degradation process have a significant influence, which indicates that R93 and R97 have a positive function on the single base degradation process (see **Supplementary Fig. 28**). We also supplemented the discussion about this appearance **in Lines 12–14 on Page 17 in the main text**.

In detail: *“The reduction of the high-state proportion and the shortening of the step length demonstrated that the stability of single base degradation was also affected.”*

5. Main text writing – in general, the authors jump quickly from one thought / tested condition to the next without explaining their logic, defining hypotheses, or summarizing conclusions. I give a few examples here:

Response: Thank this reviewer very much for the precious suggestions. We have thoroughly revised the manuscript carefully and complemented the discussions and descriptions in the deficient positions.

- For example, page 7 line 17-22 is very unclear to me as a non-expert, not explained in the methods and I cannot see what the motivation is.

Response: Thanks a lot for the useful comments on improving the accessibility of our manuscript. We have supplied the formula of the angular velocity **in the methods in Lines 6–8 on Page 24** and discussed the intention of this portion **in Lines 3–5 on Page 8 in the main text**.

In detail: *“In order to investigate the dynamic change difference of the PNPase between the RNA analog binding process and the RNA analog releasing process, we calculated the angular velocities (ω) of two adjacent data points.”*

“Angular velocity calculation.

$\omega = \arctan((y_2 - y_1)/(x_2 - x_1))$. x_1 and y_1 were coordinate values of the first data points; x_2 and y_2 were coordinate values of the second data points.”

- Repeatedly dwell time changes are interpreted on a mechanistic basis with specific molecular interactions being affected. This is especially apparent on page 13, line 3-16. The authors correctly state that they “speculate” there, but this speculation then leads to the abstract with the words “a binding event [...] was discovered for the first time”. There is no further experimental or analytical follow-up that justifies the claim in the abstract.

Response: Thanks a lot for the constructive suggestions. We sincerely apologize for our inadequate work. We have complimented the additional measurements of the mutant PNPase, where R93 and R97 were mutated to G93 and G97. The electrical measurement trajectory was shown **in**

Supplementary Fig. 28. The fitting function of the intermediate state and high state presents a single exponential decay, which indicates there was a process lost. In addition, the fingerprints of the intermediate state stemming from the electrical detecting data of the four homogeneous sequences showed no obvious differences. Furthermore, the step length of the degradation process of the mutant PNPase was found significantly less than 30, which consistently indicated that R93 and R97 play a key role in stabilizing the single-base degradation process. These results suggested that some complexation actions indeed emerged at the sites of R93 and R97. We also detailly discussed this phenomenon **in Lines 3–20 on Page 17 in the main text.**

“In order to investigate whether R93 and R97 amino acids participate in the coordination with nucleosides, we mutated R93 and R97 into G93 and G97, which do not have the structure to form hydrogen bonds. The real-time degradation process of the mutant PNPase for poly(A)₃₀ was monitored under the same experimental condition (see Supplementary Fig. 28a). Through the fitting for the intermediate state and high state, we found that both states presented a single exponential decay process, which indicated that there was a process lost after mutation (see Supplementary Fig. 28b). The dwell time of the intermediate state ($\tau = 78.84 \pm 4.17$ ms) was augmented in contrast to the value ($\tau_1 = 46.33 \pm 3.19$ ms, $\tau_2 = 8.86 \pm 0.25$ ms) before mutation, showing that the degradation process was delayed after mutation. The reduction of the high-state proportion and the shortening of the step length demonstrated that the stability of single base degradation was also affected (see Supplementary Fig. 28b). After the analysis of the electrical measurement data for the four types of the homogenous sequence, the fingerprints of the intermediate state were presented in Supplementary Fig. 28c. There was not an obvious difference in the fingerprints of the four types of homogenous sequences. In general, these results manifested that R93 and R97 participated in the coordination with the nucleosides of the RNA analog and played a key role in maintaining the stability of a single base degradation.”

- Page 9, line 19 “show” wrongly used – maybe “could indicate” ok. To use “show” further evidence is needed, e.g. by modifying the potential binding sites genetically and testing the effect in this single molecule assay

Response: Thanks a lot once again for the important suggestion. We have changed “show” to “could indicate” **in Line 8 on Page 10 in the main text** for the consideration of the preciseness of the manuscript.

- Page 13 line 20 citation is missing, and why was this analysis done in this context not given.

Response: Thanks a lot for the valuable comments. We have added the associated citation in the main text as follows: Ref. 47: “*Kinetics of RNA Degradation by Specific Base Catalysis of Transesterification Involving the 2'-Hydroxyl Group*. *J. Am. Chem. Soc.* **121**, 5364–5372 (1999)”. Through the calculation of hydrolysis reaction activation energy (E_a) based on the Arrhenius formula, k ($1/\tau_1$) as the hydrolysis rate constants, we obtained the E_a , which is consistent with the ensemble average values for ester hydrolysis. This proved that τ_1 represented the hydrolysis reaction duration. In addition, we have changed the position of this portion into the place of **Lines 8-13 on Page 14 in the main text** for the logicity of the text, which carries on the discussions about the assignment of τ_1 and τ_2 .

In detail: “Based on the Arrhenius formula, the activation energies (E_a) of hydrolysis with Mg^{2+} (87.68 ± 2.05 kJ•mol⁻¹), Mn^{2+} (105.24 ± 4.21 kJ•mol⁻¹), and D_2O (142.17 ± 3.23 kJ•mol⁻¹) were

obtained through linear fitting between $1000/T$ and k (equal to $1/\tau_1$) (Fig. 3f), which were consistent with the ensemble average values for ester hydrolysis⁴⁷. This result also proved that τ_1 was the dwell time of the hydrolysis reaction.”

[47] Li, Y. & Breaker, R. R. Kinetics of RNA degradation by specific base catalysis of transesterification involving the 2'-hydroxyl group. *J. Am. Chem. Soc.* **121**, 5364–5372 (1999).

- Thermodynamic and kinetic analysis are not comprehensive to a non-expert

Response: Thanks a lot for the meticulous consideration of this point. We have added the detailed calculation process in the method of the main text. The thermodynamic and kinetic analyses were based on the collective events of the individual behavior of a single molecule. The time interval, where data points emerged consecutively in one structure, represents the instantaneous concentration. The value is equal to the reciprocal of the occurrence number per unit of time. The reciprocal of the dwell time (hydrolysis reaction) could represent the instantaneous hydrolysis velocity constant. The analysis of the thermodynamics and kinetics was referred to the following works, such as “real-time measurement of protein–protein interactions at single-molecule resolution using a biological nanopore”, “Single-molecule Taq DNA polymerase dynamics”, and “Kinetics of RNA Degradation by Specific Base Catalysis of Transesterification Involving the 2'-Hydroxyl Group”. We have improved the description about these sections in the method of **the main text in Lines 14–19 and 22 on Page 24 as well as in Lines 1–2 on Page 25** and supplied the citations at the same time.

In detail: “As to a first-order dynamic process, K_T is the ratio of the concentration belonging to the two structures. The time interval, where data points emerged consecutively in one structure, was chosen as the instantaneous concentration, which is equal to the reciprocal of the occurrence number per unit of time ($1/p$). Therefore, K_T is equal to the value of the ratio of $1/p$ (bound) to $1/p$ (free), where p is the occurrence proportion of each structure.”

“ k is the hydrolysis reaction rate constant equal to the value of $1/\tau$ (hydrolysis reaction); and T is the thermodynamic temperature. E_a can be obtained from the slope of the linear fitting function to $\ln(k)$ and $1/RT$ ^{17,29,50,57,58}.”

[57] Thakur, A. K. & Movileanu, L. Real-time measurement of protein–protein interactions at single-molecule resolution using a biological nanopore. *Nat. Biotechnol.* **37**, 96–101 (2019).

[58] Turvey, M. W. & Gabriel, K. N. Single–molecule Taq DNA polymerase dynamics. *Adv. Sci.* **8**, eab13522 (2022).

- Unclear what are 150 or 20 sample volumes – replicates, same PNPase and 20 RNAs...?

Response: Thanks a lot for the precious suggestion on improving the accessibility of our work. We have improved the descriptions of this portion **in Lines 20–22 on Page 18 as well as in Lines 1–10 on Page 19 in the main text** and supplemented the detailed analysis routine in the method of **the main text in Lines 5–12 on Page 25**. In detail, the sequencing analysis data were chosen from the electrical measurement data of one PNPase. Firstly, we extracted 150 special data pieces and everyone must include a complete 30-step degradation process (i.e. 150 RNA degradation data). Secondly, relying on 150 couple values (dwell time & current) at every step, we plotted the fingerprint images of every site in the sequence. Then, 30 fingerprints were put together ordinarily to obtain a collection. Relying on the contrast with the reference fingerprints obtained from the homogenous sequence, we could distinguish several nucleoside information in the sequence. Finally, in the same way, we could obtain 10 collections from the bulk data. Through the information overlay of nucleoside sites from 10 collections, we could be competent to identify the information of 24

sites in the artificially designed sequence (30 bp in length) and 19 sites in a *mccA* gene sequence. In detail: “Then, given the heterogeneity of the sequence, we analyzed the dwell times and current values of the intermediate states, taking into account 150 pieces of special data (i.e. 150 RNA degradation data), where each data sample involved 30 single-base degradation steps to obtain a collection involving fingerprints of every step from the 5' terminus of the sequence (Fig. 5d). Supplementary Figs. 29–38 show 10 groups of different map collections where everyone reflects the detailed fingerprint maps of 30 sites in the sequence. Relying on comparison with the fingerprint maps ascribed to the four types of nucleosides (Fig. 4b right), nucleoside information for most sites in the 5'-CGAUCUUCAUUGCCAAGCGGCUAGCUAAA-3' sequence could be identified—indeed, through ordinal contrast of these fingerprint profiles, 24 nucleosides could be identified effectively (corresponding to about 80.00 % accuracy in the 30-nucleoside sequence) (Fig. 5e and Supplementary Fig. 39).”

In detail: “Identification of the nucleotide. 150 couple values (dwell time & current) at every step were integrated to plot a fingerprint and 30 fingerprints were put together ordinally from the 5' terminus of the sequence to obtain a collection. Relying on the contrast with the reference fingerprints obtained from the homogenous sequence, we could distinguish several nucleoside information in the sequence. finally, in the same way, we could obtain 10 collections from the bulk data. Through the information overlay of nucleoside sites from 10 collections, we could be competent to identify the information of 24 sites in the artificially designed sequence and 19 sites in a *mccA* gene sequence.”

- The last results section on the feasibility of the SiNW device for sequencing is very unclear to me, nor do I understand what 5d and 5e show as results.

Response: We apologize for our scant descriptions and thanks a lot for pointing out our shortcomings. We have carefully revised the section about the sequencing strategy by the SiNW device in **Lines 20–22 on Page 18 as well as in Lines 1–10 on Page 19 in the main text** and supplemented the detailed analysis routine in the method of **the main text in Lines 5–12 on Page 25** and changed the type of Figure 5e into the seq logo for a better acknowledgement of the identification result. Figure 5e presented the identification probability of every site to the artificially designed sequence (top panel) and a *mccA* gene sequence (bottom panel) from 10 collections. Figure 5d illustrated the conductance value and dwell time distribution of the twenty 30-step degradation processes which could reflect the stability and reliability of the detected data. We have improved the discussion about this figure in **Lines 16–20 on Page 18 in the main text**.

In detail: “The circus image of the special data (20 samples), where each sample involved 30 single-base degradation steps, is shown in Fig. 5c. It includes a scatter distribution of the current in the middle circle and a distribution of the dwell time in the inner circle. The centralized distribution of the current value and the dwell time reflected the stability and reliability of the detected data.”

- From what I understand from this section, several rounds of analysis of the same sequence are needed to identify the nucleotide sequence reliably – how would this be compatible with sequencing real transcripts that are present in few copy numbers without amplification and in a mixed pool?

Response: Thanks a lot for the precious comments. In this work, we relied on a single molecule PNPase to obtain the bulk sequencing data. For a single-molecule measurement strategy, the substrate concentration needed is very low. The ultimate sensitivity limit is also the distinct

advantage of the single molecule measurement method.

6. Methods are too brief to allow for reproduction, e.g. it would be good to have a precise explanation how dwell times are calculated, as these form the basis for many aspects of the manuscript.

Response: Thanks a lot for the constructive advice on the improvement of our manuscript. We have complemented the process of how the dwell times were obtained from the electrical measurement data through the QUB software in the method of **the main text in Lines 19–22 on Page 23 and in Lines 1–3 on Page 24.**

In detail: “In detail, relying on the difference in the conductance of each state, each state was sorted out and fitted to the continuous transformation relationship. Based on the temporal resolution between two adjacent dots, the dwell times of each state were obtained. Finally, the values of the dwell time and current could be extracted as events. Origin 2019b was then used to import and analyze the extracted data. The data of the dwell time were sorted out into 7 intervals and plotted in a histogram, where the dwell time was in the x coordinate and the count was in the y coordinate.”

We would like to take this opportunity to thank this reviewer very much for all the time involved and this great opportunity for us to improve the manuscript. We hope this reviewer will find this revised version satisfactory.

Sincerely,

The Authors

-----End of Reply to Reviewer #3-----

REVIEWER COMMENTS

Reviewer #1 (Remarks to the Author):

The authors have adequately addressed my questions and I recommend publication.

Reviewer #2 (Remarks to the Author):

In this revised manuscript, the authors present an innovative and technically sophisticated approach for real-time analysis (sequencing) of RNA using a genetically engineered derivative of polynucleotide phosphorylase (PNPase) that is coupled to a semiconductor (SiNW FET detector) device. Through a series of careful analyses, they provide evidence that these nano-scale sensors can record the degradation of an engaged RNA molecule on a millisecond timescale and with single nucleotide resolution.

The potential for the development of this sensor into a new single-molecule RNA sequencing platform is exciting and a criticism raised in the previous review was the lack of validation using real mRNAs. Even as proof of concept, this seems a critical benchmark for any new sequencing platform. In the revision, the authors sequence an RNA corresponding to 24 nucleotides of the *E. coli* *mccA* mRNA encoding a seven-residue peptide. It is not clear why this exceptionally short mRNA was chosen instead of a more conventional mRNA. The average length in *E. coli* is about 1000 nucleotides and is even greater in eukaryotes. As such, this new data provides only a marginal advance on the original 30-nucleotide artificial RNA sequence presented in the original submission. To their credit, the authors do report the successful sequencing of a mixture of these two RNAs (both chemically synthesized), which is a step closer to real-world applications. Importantly, the two test RNAs yielded approximately 80% of the expected sequence, which raises the concern that this fidelity might be much less when applied to the considerably more complex mixtures of longer RNAs found in living cells. In their response, the authors are evasive on whether or not they have actually tested longer RNAs.

Minor suggestions.

The abstract/summary and opening paragraphs of the introduction begin with the biological importance of terminal processing but this is not explored in the present work. Perhaps this is best omitted?

Page 20 line 1. If these are RNAs shouldn't the units be nucleotides and not base pairs?

Page 2 line 14. Is it warranted/necessary to state the percentage to two decimal places (80.00%)? Wouldn't 80% suffice?

Page 20 line 11. Unclear what 'universal methodology' means in this context. Are the authors suggesting a similar interface could be developed using other nucleases? If so, perhaps this could be expanded upon.

Page 20 line 14. Need to define 'CMOS-compatible playground'.

Note: It is very unhelpful to have the line numbering for individual pages rather than continuous.

Reviewer #3 (Remarks to the Author):

The points I raised were all addressed by the authors. The manuscript improved with the modified text, figures and the addition of mutant data targeting the two arginine residues that were

hypothesized to be responsible for the nucleotide-specific signal.

Panel 5e is, as the authors intended, more comprehensive, but please label the two sequences and their direction.

p8 line 15 "...was assumed to BE an unsuccessful..."

Response letter

General Reply:

We sincerely thank all the reviewers very much for their precious time involved in reviewing the manuscript and the valuable feedbacks that we have accorded to improve the quality of our manuscript. The reviewers' comments are laid out below in black font and specific concerns have been numbered. Our responses are given in blue font and changes/additions in the manuscript are given in the italic text. The places of changes/additions for the manuscript are guided with bold font in the response letter and highlighted in the revised version. We thank all the reviewers very much again for the recognition of our complementary work and hope they will be satisfied with this revised version.

Reviewer #1 (Remarks to the Author):

The authors have adequately addressed my questions and I recommend publication.

Response: We sincerely thank this reviewer very much for his/her time involved in reviewing the revised manuscript and the recognition of our complementary work.

-----End of Reply to Reviewer #1-----

Reviewer #2 (Remarks to the Author):

“In this revised manuscript, the authors present an innovative and technically sophisticated approach for real-time analysis (sequencing) of RNA using a genetically engineered derivative of polynucleotide phosphorylase (PNPase) that is coupled to a semiconductor (SiNW FET detector) device. Through a series of careful analyses, they provide evidence that these nano-scale sensors can record the degradation of an engaged RNA molecule on a millisecond timescale and with single nucleotide resolution.”

Response: We sincerely thank this reviewer very much for his/her time involved in reviewing the revised manuscript and the recognition for our complementary work. His/her previous professional suggestions have helped us significantly improve the quality of the manuscript. We appreciate his/her clear and detailed feedback once again and hope that the explanations have fully addressed all of his/her concerns on this occasion.

In the remainder of this section, we will discuss each of the comments along with our corresponding responses. The relevant revised contents in the main text are provided below in italic font for the quick reference.

1. The potential for the development of this sensor into a new single-molecule RNA sequencing platform is exciting and a criticism raised in the previous review was the lack of validation using real mRNAs. Even as proof of concept, this seems a critical benchmark for any new sequencing platform. In the revision, the authors sequence an RNA corresponding to 24 nucleotides of the *E. coli* *mccA* mRNA encoding a seven-residue peptide. It is not clear why this exceptionally short mRNA was chosen instead of a more conventional mRNA. The average length in *E. coli* is about 1000 nucleotides and is even greater in eukaryotes. As such, this new data provides only a marginal advance on the original 30-nucleotide artificial RNA sequence presented in the original submission. To their credit, the authors do report the successful sequencing of a mixture of these two RNAs (both chemically synthesized), which is a step closer to real-world applications. Importantly, the two test RNAs yielded approximately 80% of the expected sequence, which raises the concern that this fidelity might be much less when applied to the considerably more complex mixtures of longer RNAs found in living cells. In their response, the authors are evasive on whether or not they have actually tested longer RNAs.

Response: We sincerely thank this reviewer very much for his/her constructive suggestion. In this work, we realized the successful identification of heterogenous short sequences. Because of the shortage of the *E. coli* PNPase natural property, which is dominated as the degradation function for the short-piece RNA substrate in the cell, the longer sequence identification has not been achieved in our present work. This biological nature is previously revealed in the reference: “*Spickler, C. & Mackie, G. A. Action of RNase II and polynucleotide phosphorylase against RNAs containing stem-loops of defined structure. J. Bacteriol. 182, 2422–2427 (2000).*” Therefore, we will try our best to conquer this limitation by integrating with more suitable enzymes, such as Ribonuclease II or other multiprotein in the future. The current limitation of the current work has been also added in the revised manuscript for a more comprehensive understanding of our current work **in the main text in Lines 446-450 of Page 21**. “*In addition, limited by the natural property of the PNPase, which is suitable for the degradation of short-piece RNAs, the current approach has not realized the identification of longer transcripts.⁵⁶ In the future, more practical enzymes and robust analysis algorithms are urgently needed in the realization of sequencing longer transcripts.*”

2.Minor suggestions.

1). The abstract/summary and opening paragraphs of the introduction begin with the biological importance of terminal processing but this is not explored in the present work. Perhaps this is best omitted?

Response: Thanks a lot for the precious suggestions. We have removed similar descriptions, such as terminal processing **in the main text in Line 22 of Page 2.**

2). Page 20 line 1. If these are RNAs shouldn't the units be nucleotides and not base pairs?

Response: Thanks a lot for the important comments. We are sincerely apologized for our carelessness and have amended these **in the main text in Lines 421 and 425 of Page 20.**

3). Page 2 line 14. Is it warranted/necessary to state the percentage to two decimal places (80.00%)? Wouldn't 80% suffice?

Response: Thanks a lot for the key suggestion. We have realized this unnecessary description and revised these **in the main text in Line 33 of Page 2, Line 407 of Page 19, and Line 440 of Page 20.**

4). Page 20 line 11. Unclear what 'universal methodology' means in this context. Are the authors suggesting a similar interface could be developed using other nucleases? If so, perhaps this could be expanded upon.

Response: Thanks a lot for the precise consideration. The technique reported in this paper is also suitable for the investigation of other enzymes, which has the similar cysteine site on the surface. Therefore, there is a certain universality for this methodology. Thank this reviewer again for his/her valuable advice.

5). Page 20 line 14. Need to define 'CMOS-compatible playground'.

Response: Thanks a lot for the important suggestion. We have added the definition of the "*Complementary Metal Oxide Semiconductor (CMOS) -compatible playground*" **in the main text in Line 35 of Page 2.**

6). Note: It is very unhelpful to have the line numbering for individual pages rather than continuous.

Response: Thank this reviewer very much for the constructive note. We have revised this in the main text accordingly.

-----End of Reply to Reviewer #2-----

Reviewer #3 (Remarks to the Author):

The points I raised were all addressed by the authors. The manuscript improved with the modified text, figures and the addition of mutant data targeting the two arginine residues that were hypothesized to be responsible for the nucleotide-specific signal.

Response: We sincerely thank this reviewer very much for his/her time involved in reviewing the revised manuscript and the recognition for our complementary work. His/her previous professional suggestions have helped us significantly improve the quality of the manuscript. We appreciate his/her clear and detailed feedback once again and hope that the explanations have fully addressed all of his/her concerns on this occasion.

In the remainder of this section, we will discuss each of the comments along with our corresponding responses.

1. Panel 5e is, as the authors intended, more comprehensive, but please label the two sequences and their direction.

Response: Thank you very much for your important comments. We have labelled the direction and the sequences **in Fig. 5**.

2. p8 line 15 "...was assumed to BE an unsuccessful..."

Response: Thanks a lot for the key suggestion. We are sorry for our mistake and have revised this problem **in the main text in Line 170 of Page 8**.

-----End of Reply to Reviewer #3-----